# Strategic selection of MDM2 inhibitors enhances the efficacy of FAK inhibition in mesothelioma based on *TP53* genotype

Xuerao Ning[1,2], Thảo Thi Thanh Nguyễn[1,3,4], Takao Morinaga[1], Yuji Tada[5], Hideaki Shimada[6], Kenzo Hiroshima[7,8,9], Naoto Yamaguchi[2], Masatoshi Tagawa[7,10]*

1 Division of Pathology and Cell Therapy, Chiba Cancer Center Research Institute, Chiba, Japan, 2 Laboratory of Molecular Cell Biology, Graduate School of Pharmaceutical Sciences, Chiba University, Chiba, Japan, 3 Department of Molecular Biology and Oncology, Graduate School of Medicine, Chiba University, Chiba, Japan, 4 Division of Medical Biotechnology, Biotechnology Center of Ho Chi Minh City, Ho Chi Minh, Vietnam, 5 Department of Pulmonary Medicine, International University of Health and Welfare, Narita, Japan, 6 Department of Surgery, Graduate School of Medicine, Toho University, Tokyo Japan, 7 Department of Advanced Biomedical Data Science, Graduate School of Medicine, Chiba University, Chiba Japan, 8 Department of Pathology, Tokyo Women's Medical University, Yachiyo Medical Center, Yachiyo, Japan, 9 Satsukidai Hospital, Sodegaura, Japan, 10 Funabashi Orthopaedic Hospital, Funabashi, Japan

* mtagawachiba@gmail.com

## Abstract

Mesothelioma has characteristic genetic changes including inactivation of *neurofibromatosis type 2* (*NF2*) and deletion of the INK4A/ARF region. Cells deficient of NF2 protein (MERLIN) depend on focal adhesion kinase (FAK) for cell adhesion and FAK inhibitors suppress the cell growth. The INK4A/ARF deletion activates MDM2 functions which ubiquitinate and degrade p53, and consequently the cellular p53 levels decrease. The deletion therefore induces loss of p53 functions although a majority of mesothelioma has wild-type *TP53* genotype. An MDM2 inhibitor which blocked the ubiquitination increased p53 levels, restored p53 functions and facilitated cell growth arrest. Moreover, FAK and p53 expressions were reciprocally regulated. We examined growth suppressive effects of a FAK inhibitor, defactinib, and MDM2 inhibitors, nutlin-3a and reactivation of p53 and induction of tumor cell apoptosis (RITA), with representative wild-type and mutated *TP53* mesothelioma and investigated molecular changes induced by the agents. We analyzed possible combinatory effects of the inhibitors and molecular changes caused by the combination. Our study showed that defactinib inhibited cell growth and induced FAK dephosphorylation irrespective of the *TP53* genotype, and that the inhibited FAK phosphorylation was not associated with MERLIN levels or with p53 up-regulation, but linked with AKT dephosphorylation. Nutlin-3a preferentially suppressed growth of wild-type *TP53* cells and augment p53 expression without DNA damage, whereas RITA-mediated p53 up-regulation was linked with the damage. A combination of defactinib and the MDM2 inhibitors showed that nutlin-3a showed synergistic/additive effects in wild-type and antagonistic effects

**Data availability statement:** All relevant data are within the manuscript and its Supporting Information files.

**Funding:** This study was supported by Grants-in-Aid from Japan Society for the Promotion of Science (JSPS KAKENHI Grant number: 21K08199) to MT and a Grant-in-aid from the Nichias Corporation to KH and MT. The fund bodies were not involved in the design of the study and collection, analysis, and interpretation of data and in writing the manuscript. We obtained a grant from Nichias Corporation. It is not a pharmaceutical company but a company making industrial products for building, automobiles and pipes (see http://www.nichias. co.jp/). The grant is as a kind of their mécénat activities, corporate social contributions, which is aimed to assist for medical research for intractable cancer treatments. We are thereby irrelevant to any employments, consultancy, patents or products in development or marketed products of the company.

**Competing interests:** The authors have declared that no competing interests exist.

**Abbreviations:** NF2: neurofibromatosis type 2; FAK: focal adhesion kinase; RITA: reactivation of p53 and induction of tumor cell apoptosis (RITA); $IC_{50}$: half maximal inhibitory concentration; CI: combination index; Fa: fractions affected; SE: standard error; VEGF: vascular endothelial growth factor.

in mutated *TP53* cells, whereas RITA retained synergistic activity in mutated *TP53* cells. These results suggest that the therapeutic success of combined FAK and MDM2 inhibition in mesothelioma depends on the precise matching of MDM2 inhibitors with the *TP53* genotypes, and highlight the need for genotype-based selection of MDM2 inhibitors.

## Introduction

Malignant mesothelioma developed in the pleural cavity is invasive and often resistant to conventional treatments. A majority of the patients received platinum-based chemotherapy in combination with pemetrexed but the prognosis remained poor with a median survival period about 12 months [1]. The patients often become resistant to the agents and a second-line drug is currently unavailable. Recent clinical studies however showed that immune checkpoint inhibitors were as effective as the first-line chemotherapeutic agents [2–4] although the inhibitors produced significant adverse effects. Nevertheless, the immune checkpoint inhibitors did not produce better therapeutic effects than chemotherapy in relapsed mesothelioma patients [4]; consequently, a novel agent is required for further improvement of the treatment strategy.

Malignant mesothelioma has 2 characteristic genetic abnormalities, inactivation of *neurofibromatosis type 2* (*NF2*) [5] and deletion of the INK4A/ARF region which includes p16[INK4A] and p14[ARF] genes [6]. The *NF2* gene encodes an ezrin-radixin-moesin-like protein, MERLIN, which functions as a tumor suppressor and mediates contact inhibition through several signaling processes [7]. A recent study also demonstrated that mesothelioma cells deficient of MERLIN expression were susceptible to an inhibitor of focal adhesion kinase (FAK), a non-receptor-type tyrosine kinase localized to focal contacts [8]. FAK plays a role in cell proliferation and the inhibitor in particular suppressed growth of MERLIN-deficient cells since the MERLIN-negative cells were more dependent on the FAK signaling for the cell adhesion and proliferation than the positive cells. These data suggested that a FAK inhibitor was a candidate for mesothelioma treatments since mesothelioma was often defective of MERLIN expression. On the other hand, a different study showed that an expression level of E-cadherin, which also mediates cell adhesion, was linked with resistance to a FAK inhibitor in MERLIN-deficient cells [9]. Several clinical studies with FAK inhibitors were conducted for mesothelioma patients [10–12], but a possible association of susceptibility to FAK inhibitors with an expression level of MERLIN or other molecules was not well understood.

The deletion of INK4A/ARF region results in defective p53 activities even though a majority of mesothelioma has the wild-type *TP53* genotype. Loss of the *p14* gene, mapped in the deleted region, augments the MDM2 function which degrades wild-type p53 through enhanced p53 ubiquitination [13]. The deletion was frequently found in clinical specimens of mesothelioma and consequently, the majority showed functional loss or deficiency of p53 actions. An inhibitor of the MDM2 function can stabilize p53 protein by blocking the degradation process and increase the level,

which activates p53-mediated pathways. A small-sized inhibitor interfering a binding between p53 and MDM2, for example nutlin-3a and reactivation of p53 and induction of tumor cell apoptosis (RITA), in fact up-regulated a p53 expression level and stimulated the p53 pathways; consequently, the inhibitor induced growth inhibition and apoptotic cell death in human tumors [14–16]. The inhibitors and a combination with the first-line agent also showed anti-tumor effects on mesothelioma [17]. Activation of the p53-mediated pathway is thereby a possible therapeutic strategy for mesothelioma with wild-type *TP53* gene. On the other hand, effects of up-regulation of mutated p53 on tumor cell growth remained unclear.

Several previous studies showed a possible interaction between FAK and p53, and a reciprocal regulation between the molecules [18,19]. Knock-down of FAK expression increased p53 levels [18] and up-regulation of p53 suppressed FAK levels [19]. In addition, MERLIN played a certain role in p53 up-regulation through inhibition of the MDM2 function [20]. These previous data collectively suggested that a combination of a FAK inhibitor and a MDM2 inhibitor augmented p53 expression levels and induced growth inhibition even in MERLIN-positive mesothelioma. In fact, a FAK inhibitor augmented growth inhibitory effects of a p53-activating chemical agent in mesothelioma with wild-type *TP53* genotype [21]. Nevertheless, a growth suppressive activity of a FAK inhibitor in terms of MERLIN expression and *TP53* genotype was not well studied, and possible combinatory effects of a FAK inhibitor and a MDM2 inhibitor with respect to the *TP53* genotype remained uncharacterized. In this study, we investigated growth suppressive activity of defactinib (also known as VS-6063), a FAK inhibitor used for clinical studies, and MDM2 inhibitors, nutlin-3a and RITA, and further examined a possible combinatory effect on mesothelioma cells with different MERLIN expression levels and *TP53* genotypes.

## Methods

### Cells and agents

Human mesothelioma cells, NCI-H28, MSTO-211H, NCI-H2052, NCI-H226 and NCI-H2452 cells, and SV40-T antigen-expressing immortalized cells of mesothelium origin, Met-5A cells, were purchased from American Type Culture Collection (Manassas, VA, USA). Mesothelioma with mutated *TP53* genotype, EHMES-1 (R273S) and JMN-1B (G245S) cells, and that with wild-type *TP53* gene, EHMES-10 cells, were provided by Dr. Hironobu Hamada (Hiroshima University, Japan) [22]. Cells were cultured with in RPMI-1640 medium supplemented with 10% fetal calf serum, and were confirmed to be negative for mycoplasma. The genotype of *TP53* was wild-type in NCI-H28, MSTO-211H, NCI-H2052, NCI-H226 and NCI-H2452 cells, but p53 protein of NCI-H2452 cells was truncated (Fig 1, S1 Table). Chemicals used in the present study,

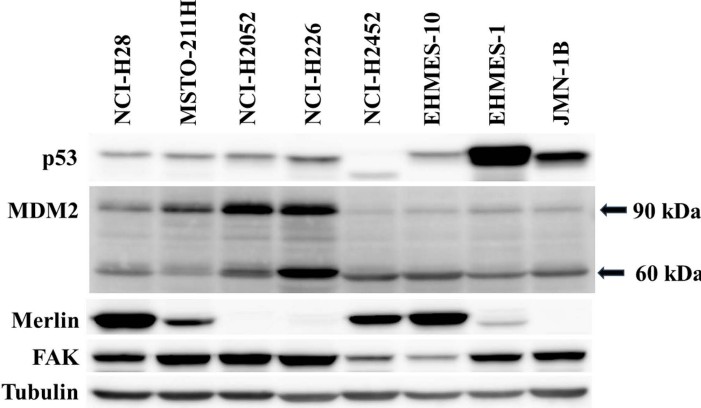

**Fig 1. Expression levels of p53, MDM2, MERLIN and FAK in mesothelioma cells with western blot analysis.** Tubulin was used as a loading control. The expression of each molecule was quantified with ImageJ software (NIH, Bethesda, MD, USA) with tubulin intensity as a normalized control (see S3 Table).

defactinib (VS-6063), nutlin-3a (S8059) and RITA (NSC 652287) were purchased from Selleck Chemicals (Houston, TX, USA).

## Viable cell detection assay

Cells seeded in 96-well plates ($1 \times 10^3$ cells per well) were treated with an agent alone or in a combination of 2 agents for 72 hrs. Cell viability was determined with a cell-counting WST kit (Wako, Osaka, Japan) with absorbance at 450 nm using a microplate reader (Model 620, Bio-Rad, Hercules, CA, USA). The relative viability was calculated based on the absorbance without any treatments, an absorbance of cells treated with an agent/(an absorbance of cells untreated – an absorbance of medium only) x 100 (WST assay). Half maximal inhibitory concentration ($IC_{50}$) values and combinatory effects were examined with the CalcuSyn software (Ver2, Biosoft, Cambridge, UK). Combination index (CI) values at respective fractions affected (Fa) points, which showed relative suppression levels of the cell viability, were calculated based on the cell viability test. CI < 1, CI = 1 and CI > 1 indicates synergistic, additive and antagonistic actions, respectively. Statistical analysis was performed with one-way analysis of variance.

## Western blot analysis

Cell lysate was subjected to sodium dodecyl sulfate-polyacrylamide gel electrophoresis. The protein was transferred to a nylon filter membrane and was hybridized with antibody against phosphorylated p53 at Ser 15 (#9284), AKT (#9272), phosphorylated AKT at Ser 473 (#9271), caspase-9 (which also detected cleaved caspase-9) (#9505), PARP (poly ADP-ribose polymerase) (which also detected cleaved PARP) (#4108), FAK (#3285), phosphorylated FAK at Tyr 397 (#3283), phosphorylated MDM2 at Ser 166 (#3521), MERLIN (D1D8, #6995), actin (#4970) (Cell Signaling, Danvers, MA, USA), phosphorylated H2AX (phosphor-2A histone family member X) at Ser 139 (#613401) (BioLegend, San Diego, CA, USA), MDM2 (#413) (Santa Cruz Biotechnology, Santa Cruz, CA, USA), p53 (Ab-6, clone DO-1, #MS-187-P0), and tubulin-α (DM1A, Thermo Fisher Scientific) followed by an appropriate second antibody (S2 Table). The membranes were developed with the ECL system (GE Healthcare, Buckinghamshire, UK). Actin and tubulin-α were used as a loading control. We chose an optimal exposure time and selected a representative blot to visualize molecular changes for each experiment.

## Statistical analysis

Statistical analyses were conducted with unpaired t-test in GraphPad Prism 6 software (GraphPad, La Jolla, CA, USA) and P-values less than 0.05 were considered as statistically significant.

## Results

### Expression of p53, MERLIN and FAK in mesothelioma cells

We examined 8 kinds of mesothelioma cells for the expression of p53, MDM2, MERLIN and FAK with western blot analysis (Fig 1, S3 Table). Mesothelioma with the wild-type *TP53* genotype (NCI-H28, MSTO-211H, NCI-H2052, NCI-H226 and EHMES-10) expressed p53 at a lower level than those with mutated *TP53* genotype (EHMES-1 and JMN-1B) since wild-type p53 was more susceptible to the ubiquitin-mediated degradation than mutated p53. NCI-H2452 cells expressed truncated p53 despite having the wild-type genotype, and consequently the p53 functions were defective. The anti-MDM2 antibody detected both full-length 90 kDa and cleaved 60 kDa molecules, and both of which molecules were able to ubiquitinate p53 [23]. Mesothelioma cells with the wild-type *TP53* genotype except NCI-H2452 and EHMES-10 cells expressed 90 kDa MDM2 at a relatively high level compared with those with mutated *TP53*. MERLIN was positive in NCI-H28, MSTO-211H, NCI-H2452, and EHMES-10 cells, and was minimally expressed in EHMES-1 cells, whereas the expression was almost undetectable in the other cells. All the mesothelioma expressed FAK, but the level was relatively low in NCI-H2452 and EHMES-10 cells compared with that of the other cells. These data indicated that an expression

level of FAK was not associated with that of MERLIN or MDM2, and was independent of *TP53* genotype. In addition, a MERLIN expression level was irrelevant to the *TP53* genotype and an MDM2 level.

## Growth inhibitory effects of defactinib on mesothelioma

We examined a growth inhibitory activity of defactinib with the colorimetric WST assay (Fig 2A). All the 8 kinds of meso-thelioma and Met-5A cells which were defective in the p53 functions due to expression of the p53-inactivating SV40-T antigen, showed similar sensitivity to defactinib. The $IC_{50}$ values were not different between mesothelioma cells with wild-type *TP53* genotype (average ± SE; 3.85 ± 0.65 μM) and those with mutated *TP53* genotype (EHMES-1 and JMN-1B) or with non-functional p53 (NCI-2452 and Met-5A) (4.52 ± 1.17) (P = 0.64). Furthermore, sensitivity to the FAK inhibitor was unrelated to the MERLIN expression levels (MERLIN-high cells: NCI-H28, MSTO-211H, NCI-H2452 and EHMES-10 $IC_{50}$ = 5.98 ± 1.00, MERLIN low or negative cells: NCI-H2052, NCI-H226, EHMES-1 and JMN-1B $IC_{50}$ = 3.47 ± 0.85) (P = 0.105) (Fig 1). We also examined growth inhibitory activity of defactinib with a dye exclusion assay, which measured viable cell numbers with a trypan blue dye, and confirm the cell growth inhibition (S1A Fig). The assay showed that defac-tinib inhibited the proliferations of all the cells tested.

We next examined expression of FAK, p53, AKT and MDM2 with western blot analysis (Fig 2B, S4 Table). Meso-thelioma cells treated with defactinib showed dose-dependent decrease of FAK phosphorylation levels at Tyr 397, an autophosphorylation site controlling the FAK activity, whereas the FAK expression levels of the treated cells remained

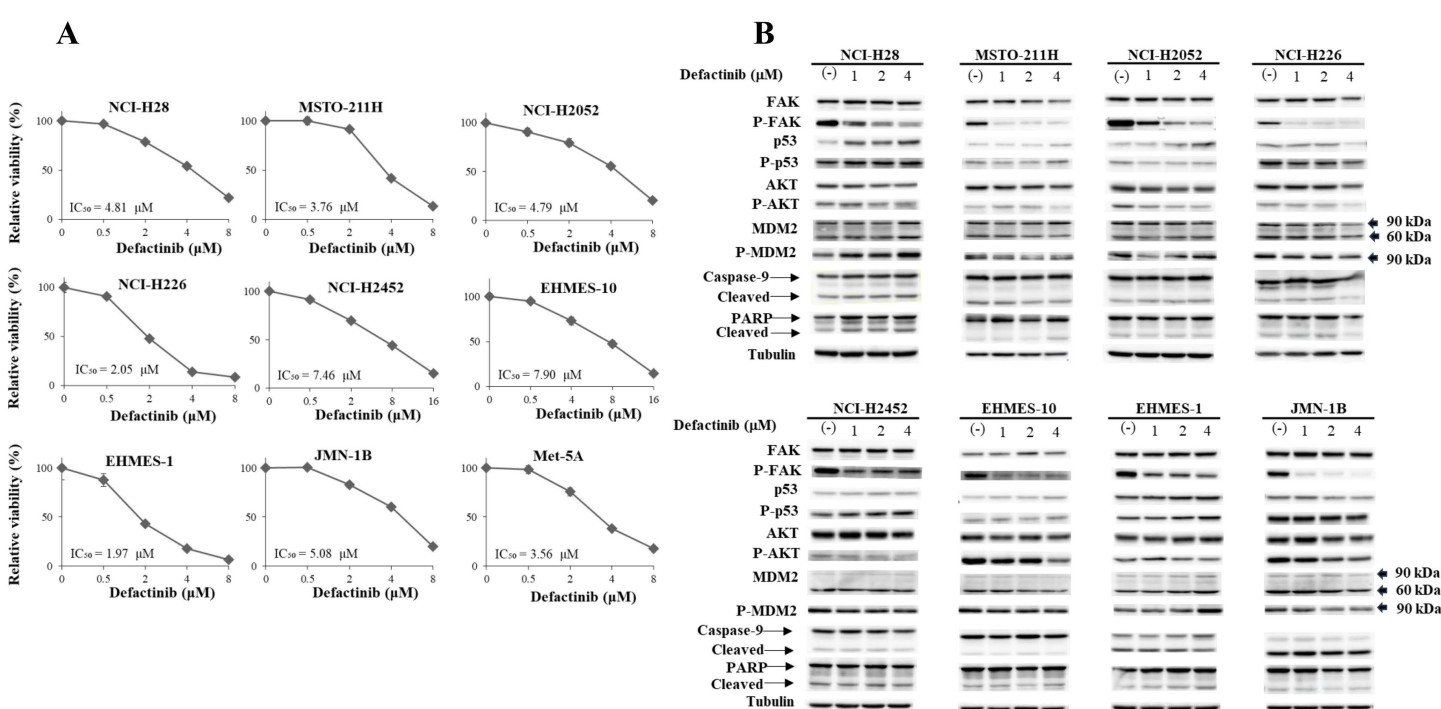

**Fig 2. Growth inhibition and molecular changes induced by defactinib.** (A) Growth inhibitory effects of defactinib on mesothelioma and Met-5A cells. Cell cells were treated with defactnib as indicated for 72 hrs and the viability was assayed with the WST assay. $IC_{50}$ values and SE bars (n = 3) are also included. (B) Expression level of the molecules associated with FAK and p53 in mesothelioma cells treated with defactinib as indicated for 24 hrs. The western blot analysis included expression levels of a phosphorylated form of FAK, p53 and AKT, and a cleaved form of caspase-9 and PARP. Tubu-lin was used as a loading control. The expression of each molecule was quantified with ImageJ software (NIH, Bethesda, MD, USA) with tubulin intensity as a normalized control (see S4 Table).

unchanged except MSTO-211H and NCI-H226 cells. The data confirmed that defactinib at the doses used in the experiments, 1−4 μM, inhibited a kinase activity of FAK. The defactinib treatment augmented expression of p53 or phosphorylated p53 at Ser 15 in NCI-H28, MSTO-211H, NCI-H2052, NCI-H2452, EHMES-10 and EHMES-1 cells, whereas the treatment decreased the expression in NCI-H226 and JMN-1B cells. Phosphorylation of p53 at Ser 15 is a maker for p53 stabilization and can be linked with increase of p53 expression, and the linkage between the phosphorylation and increased p53 level was detected in NCI-H28, MSTO-211H, NCI-H2452 and EHMES-1 cells. These data indicated that FAK inhibition was not always associated with increase of p53 levels or the phosphorylation levels irrespective of the *TP53* genotype. AKT (protein kinase B) is responsible for many cellular processes and located down-stream of receptor-type tyrosine kinases. We found that phosphorylated AKT at Ser 473 levels, which was a marker of AKT activation, and less significantly AKT levels were down-regulated in all the cells by the defactinib treatments. The decreased AKT phosphorylation was probably attributable to the defactinib-induced growth retardation and was consequently correlated with down-regulation of FAK phosphorylation. We also found that that MDM2 levels remained unchanged but the phosphorylation was differently regulated among the cells tested; increased phosphorylation in NCI-H28 and EHMES-1 cells, and decreased in EHMES-10 and JMN-1B cells. Regulation of the MDM2 phosphorylation was thus not directly linked with inhibited FAK phosphorylation, *TP53* genotype, p53 or MERLIN expression levels. Cleavages of caspase-9 was slightly enhanced in NCI-H28 cells and cleaved PARP was also minimally up-regulated in NCI-H28 and EHMES-1 cells, whereas other cells did not show the enhanced cleavages. These data collectively indicated that suppressed FAK activity by the defactinib was associated with inhibition of AKT phosphorylation, but not directly linked with phosphorylation of p53 and MDM2, or *TP53* genotype. Moreover, the defactinib treatment did not induce apoptotic cell death in most of the cells tested.

## Growth inhibitory effects of nutlin-3a were linked with p53 functional type

We next examined sensitivity of mesothelioma and Met-5A cells to nutlin-3a, an MDM2 inhibitor, with the WST assay and calculated the $IC_{50}$ values (Fig 3A). Cells with the wild-type *TP53* genotype excluding EHMES-10, were more susceptible to nutlin-3a ($IC_{50}$ = 3.62 ± 1.04 μM) than those with mutated *TP53* and non-functional p53 (33.5 ± 2.80) (P < 0.01). These data showed that nutlin-3a sensitivity of mesothelioma cells was correlated with *TP53* genotype and the p53 functional status except EHMES-10 cells. A mechanism of the nutlin-3a resistance in EHMES-10 cells is currently unknown, but the $IC_{50}$ value of the cells was relatively close to that of mutated *TP53* or non-functional p53 cells. The cells were also resistant to RITA-mediated growth inhibition (see below). These data suggested that EHMES-10 cells were insusceptible to growth inhibition or cell death probably because the p53 down-stream pathway was disturbed, in which reduced p21 induction limits cell cycle arrest.

We also examined expression and phosphorylation levels of p53 and FAK in mesothelioma cells treated with nutlin-3a (Fig 3B, S5 Table). We used representative mesothelioma cells in terms of the *TP53* genotype, NCI-H28, MSTO-211H and NCI-H226 as for wild-type *TP53* cells, and EHMES-1 and JMN-1B as for mutated *TP53* cells. All the wild-type *TP53* cells showed up-regulation of p53 levels and the phosphorylation. MSTO-211H and NCI-H226 cells down-regulated FAK phosphorylation levels but NCI-H28 cells remained unchanged in the FAK phosphorylation. Levels of phosphorylated H2AX at Se 139, a marker for DNA damage, were variable with nutlin-3a doses in the wild-type *TP53* cells and the expression levels were not dose-dependent. These data suggested that p53 up-regulation by nutlin-3a was not due to DNA damage but probably due to inhibition of the MDM2 activity [13]. Mesothelioma with mutated *TP53* showed different responses. Nutlin-3a-treated JMN-1B and EHMES-1cells increased the p53 level and the phosphorylated level, respectively. FAK and the phosphorylation levels were down-regulated in JMN-1B cells but those were unchanged in EHMES-1 cells. Phosphorylated H2AX levels remained constant in both mutated *TP53* cells. These data collectively indicated that nutrin-3a augmented p53 expression in wild-type and mutated *TP53* cells with a different manner and that the association of the p53 increase with inhibited FAK phosphorylation was dependent on cells

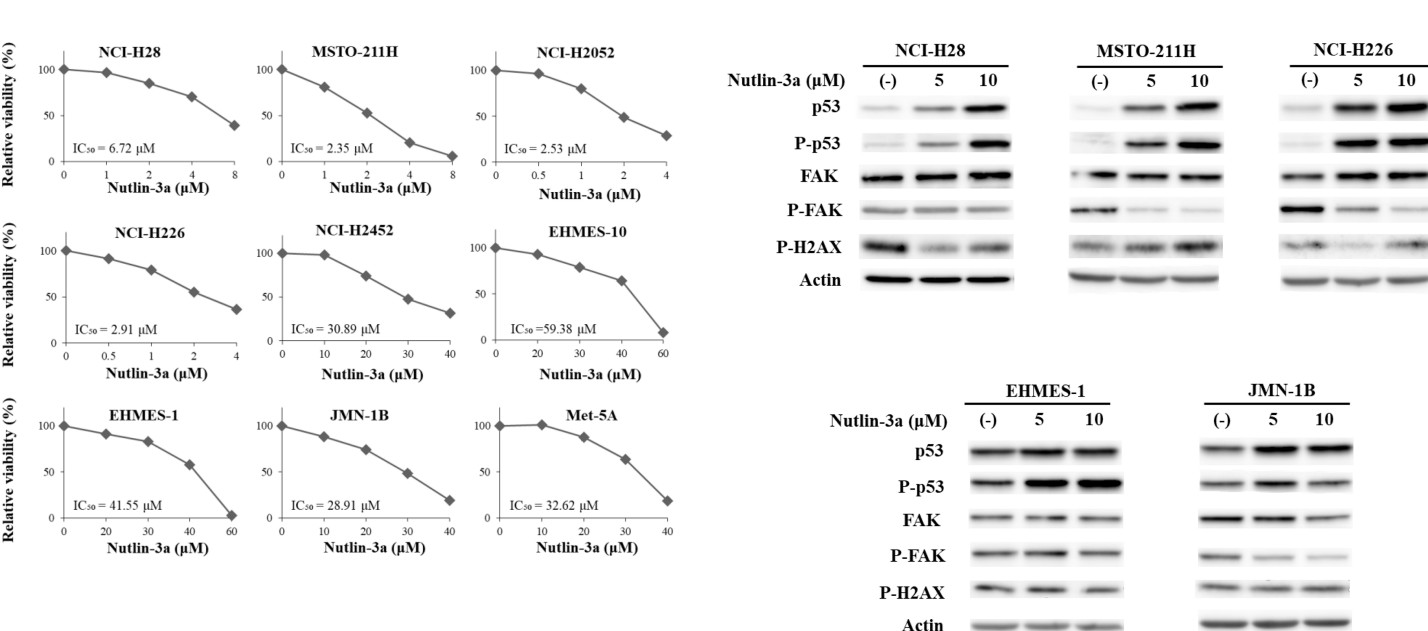

**Fig 3. Growth inhibition and molecular changes induced by nutlin-3a.** (A) Growth inhibitory effects of nutlin-3a on mesothelioma and Met-5A cells. Cell cells were treated with nutlin-3a as indicated for 72 hrs and the viability was assayed with the WST assay. $IC_{50}$ values and SE bars (n = 3) are also included. (B) Expression level of the molecules associated with FAK and p53 in mesothelioma cells treated with nultin-3a as indicated for 24 hrs. The western blot analysis also included expression levels of a phosphorylated form of FAK, p53 and H2AX. Actin was used as a loading control. For phosphorylated H2AX, the blots were exposed for a longer period to ensure the detection of basal signals in untreated cells. The basal intensity therefore differed from that of the RITA experiments in Fig 4B where shorter exposures were used. The expression of each molecule was quantified with ImageJ software (NIH, Bethesda, MD, USA) with actin intensity as a normalized control (see S5 Table).

irrespective of the *TP53* genotype. In addition, nutlin-3a-mediated p53 increase and FAK dephosphorylation were not linked with MERLIN expression levels.

### Growth inhibitory effects of RITA were irrelevant to *TP53* genotype

We examined growth inhibitory effects of RITA, another type of the MDM2 inhibitors, with the WST assay (Fig 4A). The $IC_{50}$ values of mesothelioma with the wild-type *TP53* genotype excluding EHMES-10 (2.50 ± 1.88 μM) were not different from those of cells with mutated *TP53* and non-functional p53 (1.67 ± 1.21) (P = 0.73). EHMES-10 cells were resistant to RITA and nutlin-3a as mentioned above. Susceptibility of mesothelioma to RITA was thus dependent on the cells tested and was not associated with the *TP53* genotype or with the p53 functional status.

We then examined p53 and FAK expression in RITA-treated cells with western blot analysis (Fig 4B, S6 Table). Expression levels of both p53 or the phosphorylation increased in the wild-type and mutated *TP53* cells, but EHMES-1 cells decreased the p53 level despite enhanced p53 phosphorylation. Phosphorylated p53 at Ser 15 was thus not always linked with augmented p53 levels in the mutated *TP53* cells as found in nutrin-3a-treated mutated *TP53* cells. All the RITA-treated cells showed decreased FAK phosphorylation and increased H2AX phosphorylation levels, with the exception of MSTO-211H cells exhibiting a minor H2AX increase. These data suggested that DNA damage responses contributed to RITA-mediated p53 up-regulation and that increased p53 phosphorylation was linked with FAK dephosphorylation irrespective of the *TP53* genotype.

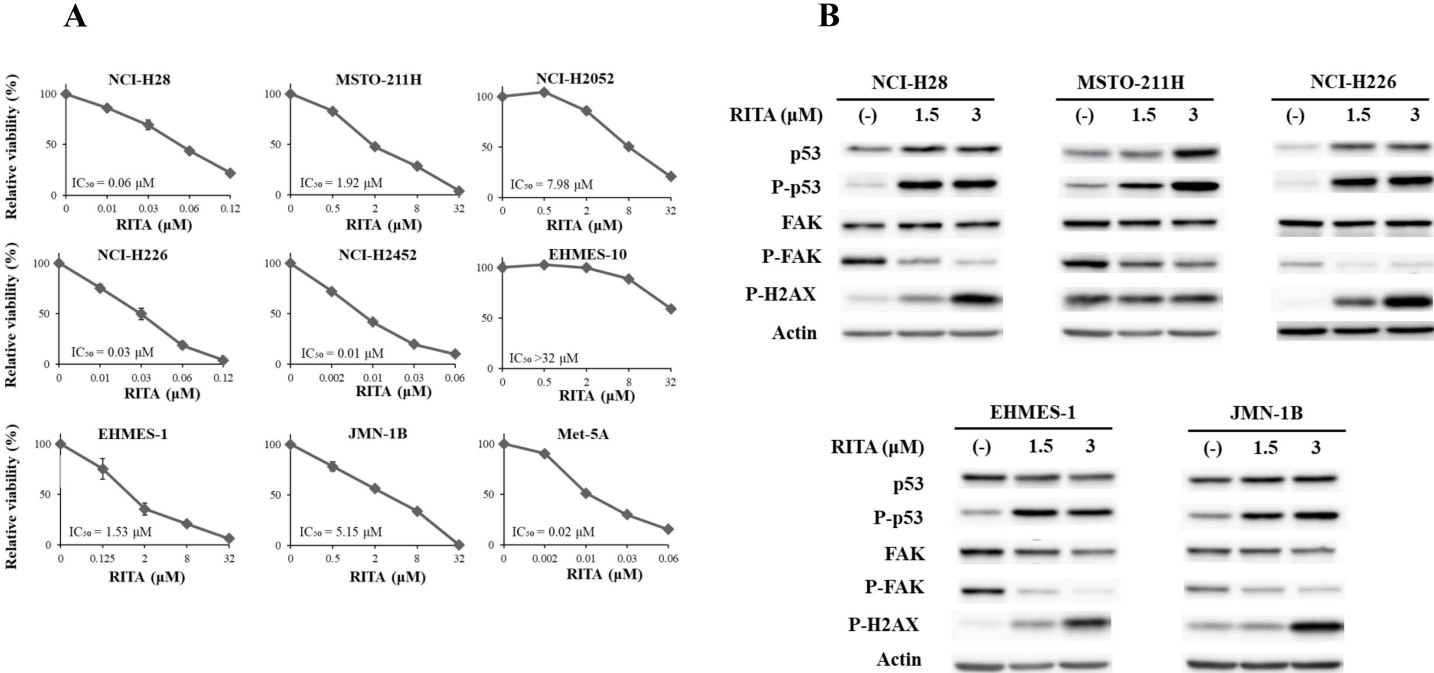

**Fig 4. Growth inhibition and molecular changes induced by RITA.** (A) Growth inhibitory effects of RITA on mesothelioma and Met-5A cells. Cell cells were treated with RITA as indicated for 72 hrs and the viability was assayed with the WST assay. $IC_{50}$ values and SE bars (n = 3) are also included. (B) Expression level of the molecules associated with FAK and p53 in mesothelioma cells treated with RITA as indicated for 24 hrs. The western blot analysis included expression levels of a phosphorylated form of FAK, p53 and H2AX. Actin was used as a loading control. For phosphorylated H2AX of MSTO-211H cells, the blot was exposed for a longer period to ensure the detection of basal signals in untreated cells. The expression of each molecule was quantified with ImageJ software (NIH, Bethesda, MD, USA) with actin intensity as a normalized control (see S6 Table).

## Combinatory effects of defactinib and MDM2 inhibitors

We investigated a possible combinatory effect produced by defactinib and the MDM2 inhibitors with the WST assay. We treated mesothelioma cells with defactinib at a concentration of about 10−30% growth inhibition together with various doses of nutlin-3a (Fig 5) or RITA (Fig 6). The defactinib treatment further enhanced nutlin-3a-induced growth suppression even in nutlin-3a-resistant cells including those with mutated *TP53* genotype and with truncated p53, and EHMES-10 cells (Fig 5A). Further investigations showed that the majority of CI values at Fa points between 0.25 and 0.75 was below or close to 1 in nutlin-3a-sensitive wild-type *TP53* mesothelioma, whereas the corresponding CI values in 2 kinds of mutated *TP53* mesothelioma cells were above 1 (Fig 5B). These data indicated that wild-type *TP53* cells sensitive to nutlin-3a showed synergistic or additive growth inhibition with the FAK inhibitor but mutated *TP53* cells rather showed antagonistic effects. In contrast, CI values of NCI-H2452 cells with truncated p53 were dependent on Fa points and those of nutlin-3a-resistant EHMES-10 were close to 1. We also examined growth inhibitory effects of defactinib in combination with RITA (Fig 6A). The defactinib treatment augmented RITA-induced growth suppression in all the cells tested. The CI values at the Fa points between 0.25 and 0.75 were below or about 1 in all the cells except NCI-H226 and NCI-H2452 cells, both of which showed most of CI values above 1 (Fig 6B). The CI value data indicated that the combination produced synergistic effects irrespective of the *TP53* genotype but the synergism was dependent on cells used. The combination of defactinib and the MDM2 inhibitors thereby achieved synergistic or additive effects in most of wild-type *TP53* cells, but the combination effects tested in mutated *TP53* or non-functional p53 cells were dependent on the inhibitor and the cells tested.

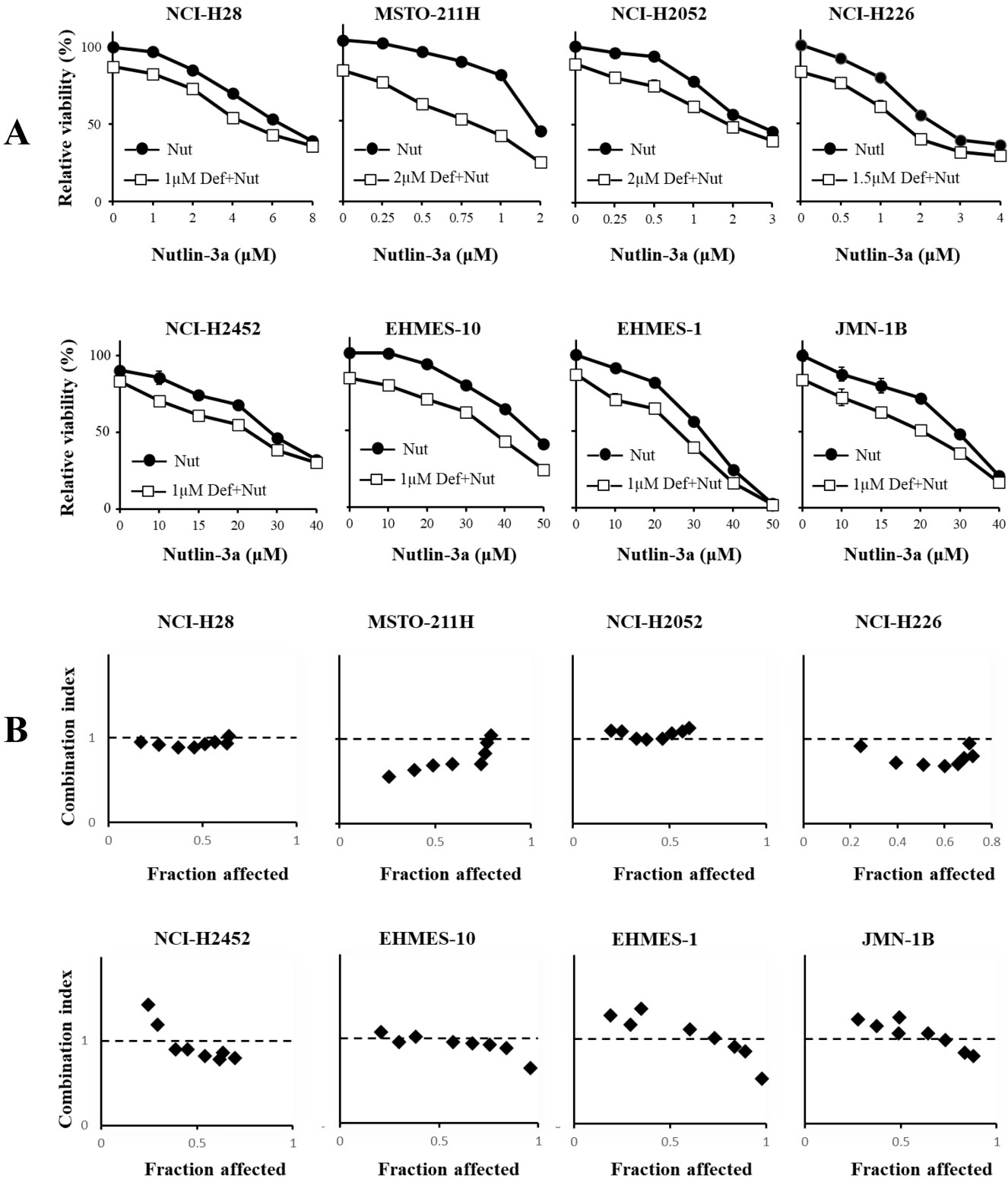

**Fig 5. Combination of defactinib and nutlin-3a.** (A) Combinatory effects of defactinib and nutlin-3a on mesothelioma. Cells were treated with defactinib and nutlin-3a as indicated for 72 hrs and the viability was assayed with the WST assay. SE bars (n = 3) are shown. (B) The combinatory effects were examined with the CalcuSyn software and the CI values at respective Fa points are shown.

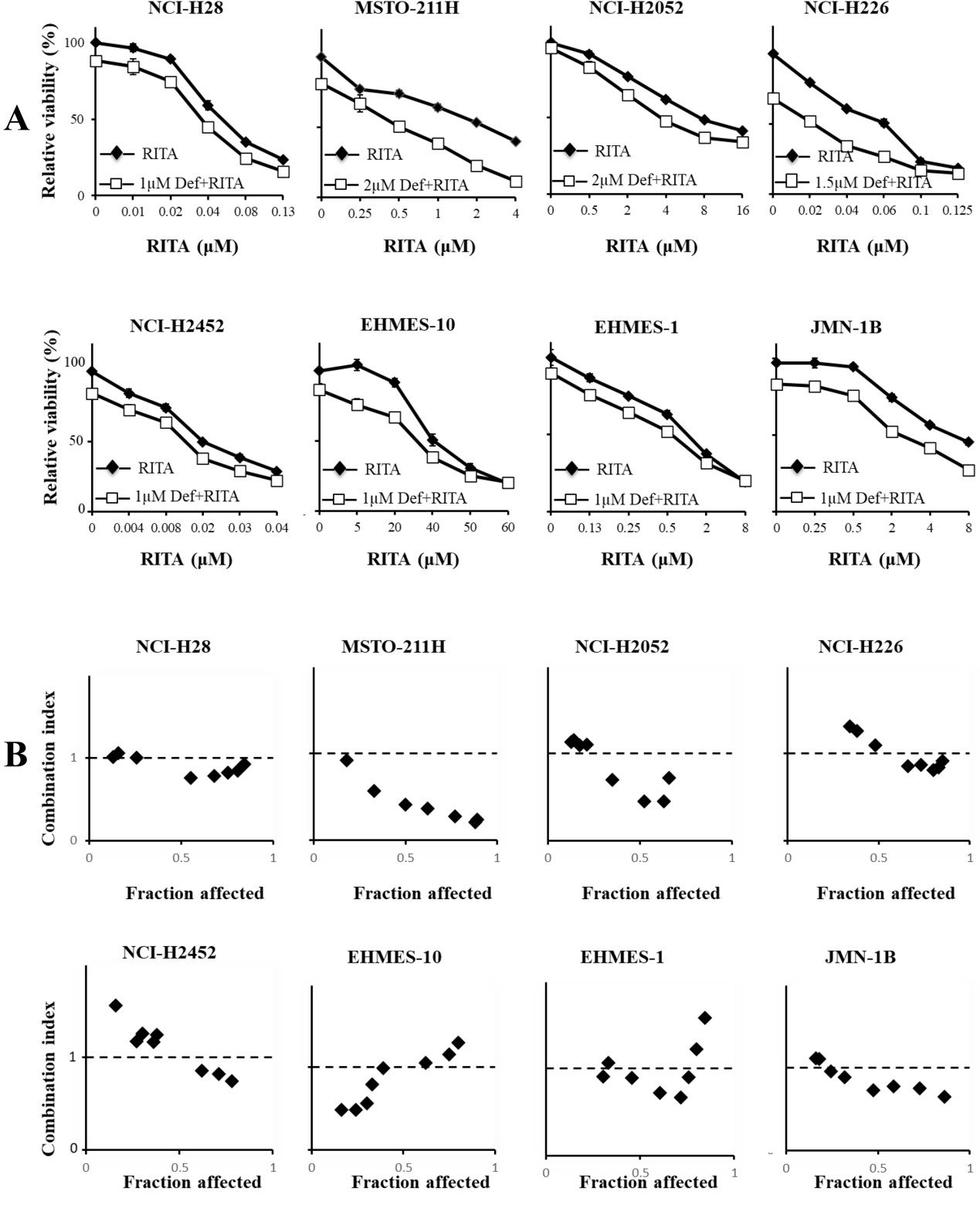

**Fig 6. Combination of defactinib and RITA.** (A) Combinatory effects of defactinib and RITA on mesothelioma. Cells were treated with defactinib and RITA as indicated for 72 hrs and the viability was assayed with the WST assay. SE bars (n = 3) are shown. (B) The combinatory effects were examined with the CalcuSyn software and the CI values at respective Fa points are shown.

We then examined molecular changes produced by the combination of defactinib and the MDM2 inhibitors in representative *TP53* wild-type and mutated cells (Fig 7, S7 Table). We treated the cells with defactinib and the MDM2 inhibitors at a concentration of a little lower than the respective IC$_{50}$ values in the growth inhibition study to evaluate synergistic or additive effects. (Fig 2A, 3A and 4A, see the Fig 7 legend). We examined RITA only for mutated *TP53* cells in the combination because the mutated cells were resistant to nutlin-3a and a high concentration of nutlin-3a could produce non-specific molecular changes. The defactinib treatment decreased FAK phosphorylation in all the cells irrespective of the *TP53* genotype, and the combination with nutlin-3a did not further enhance the FAK dephosphorylation in *TP53* wild-type cells. A combination with RITA however further down-regulated FAK phosphorylation in NCI-H28 and *TP53* mutated cells, and the FAK phosphorylation level in these cells was lower in the combination than in a treatment with RITA or defactinib

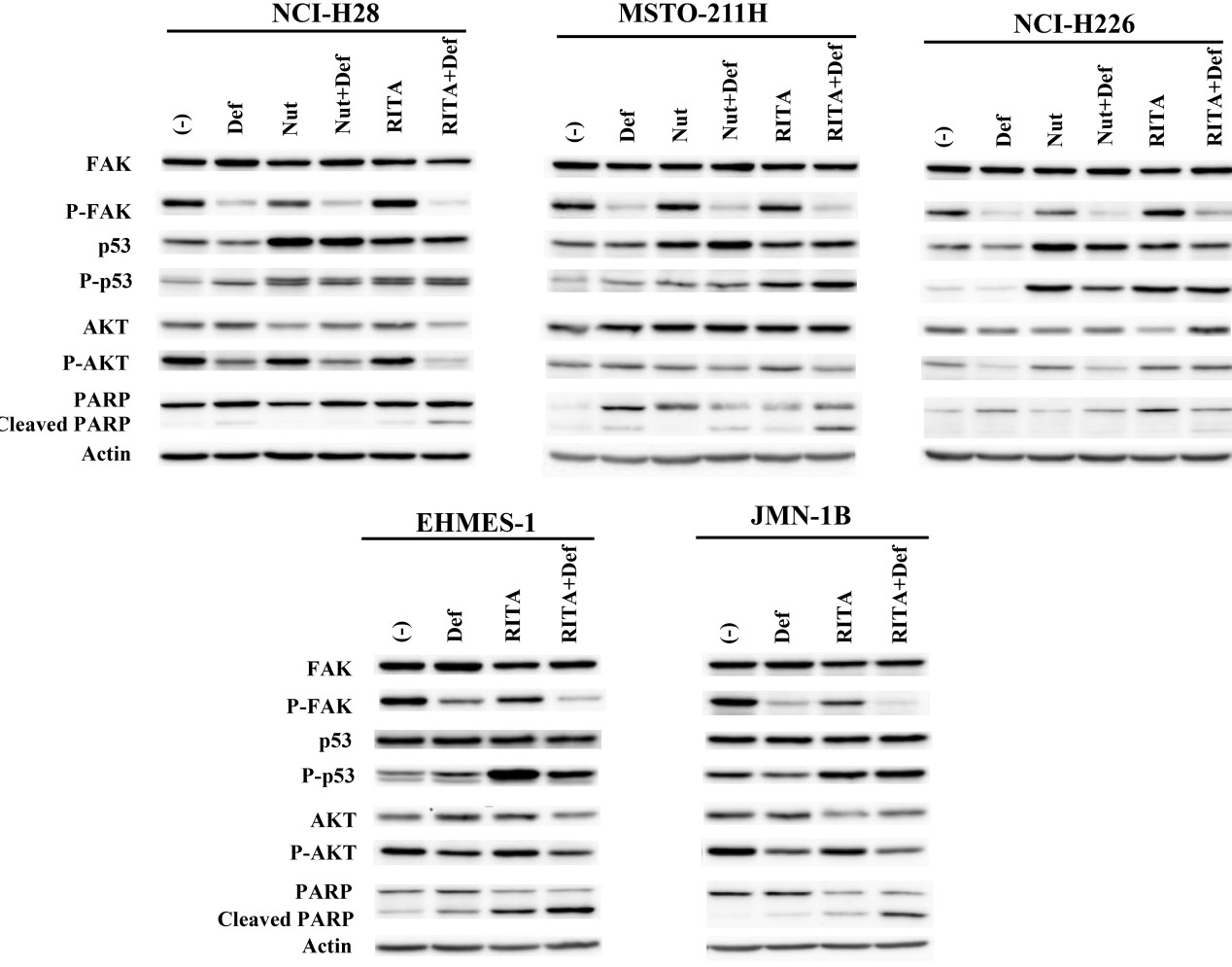

**Fig 7. Expression level of the molecules associated with FAK and p53 in mesothelioma cells treated with defactinib and/or the MDM2 inhibitor, nutlin-3a or RITA.** Cells were treated as follows for 24 hrs and the cell lysate were subjected to western blot analysis, which included expression levels of a phosphorylated form of FAK, p53 and AKT, and a cleaved form of PARP. We selected concentrations lower than the IC$_{50}$ values of each single agent to detect the molecular changes induced by the combination. Concentrations of defactinib, nutrin-3a and RITA used were as follows. NCI-H28: 2 µM defactinib, 4 µM nutlin-3a, 0.04 µM RITA; MSTO-211H: 2 µM defactinib, 1 µM nutlin-3a, 1 µM RITA; NCI-H226: 2 µM defactinib, 2 µM nutlin-3a, 0.02 µM RITA; EHMES-1: 2 µM defactinib, 0.25 µM RITA; JMN-1B: 2 µM defactinib, 1.5 µM RITA. Actin was used as a loading control. The expression of each molecule was quantified with ImageJ software (NIH, Bethesda, MD, USA) with actin intensity as a normalized control (see S7 Table).

alone. The synergistic combinatory effects of defactinib and RITA on growth inhibition was therefore associated with decreased FAK activity in some mesothelioma cells, but the effects of defactinib and nutlin-3a were not directly relevant to FAK dephosphorylation in wild-type *TP53* cells. Nutrin-3a-treated wild-type *TP53* cells showed up-regulated p53 and the phosphorylation but the combination with defactinib did not further enhance the expressions except the p53 level in MSTO-211H cells. NCI-H226 cells showed that the combination with defactinib rather decreased the p53 expression. RITA-treated wild-type *TP53* cells increased p53 and the phosphorylation, but additional defactinib did not further augment the expression levels. RITA-treated mutated *TP53* cells up-regulated p53 phosphorylation, but the combination with defactinib showed different responses; decreased and unchanged levels in EHMES-1 and JMN-1B cells, respectively. The MDM2 inhibitors augmented p53 and the p53 phosphorylation, but the combination of defactinib did not always contribute to the p53 up-regulation; consequently, the p53 regulations by MDM2 inhibitors were not influenced by inhibited FAK phosphorylation.

Defactinib treatments decreased AKT phosphorylation in all the cells except MSTO-211H cells, but additional nutlin-3a did not further down-regulate the phosphorylation in wild-type *TP53* cells except MSTO-211H cells. RITA did not influence AKT phosphorylation in wild-type or mutated *TP53* cells except JMN-1B cells. The combination with defactinib and RITA further decreased the phosphorylation in NCI-H28 and slightly in mutated *TP53* cells. PARP molecules were cleaved by defactinib or nutlin-3a, and the combination did not enhance the cleavages. Combinatory treatments of RITA and defactinib increased PARP cleavage more than a treatment with RITA or defactinib alone regardless of the *TP53* genotype except in NCI-H226 cells. These data suggested that synergistic growth inhibitory effects in the combination of defactinib and nutlin-3a were not directly associated with FAK dephosphorylation, p53 phosphorylation or AKT dephosphorylation levels in wild-type *TP53* cells. The combination of defactinib and RITA achieved synergistic effects in the majority of wild-type and mutated *TP53* cells, and the effects were in general associated with down-regulated FAK and AKT phosphorylation, and apoptotic cell death induced.

## Discussion

We showed in this study that susceptibility of mesothelioma to defactinib was not associated with MERLIN expression or the *TP53* genotype, and demonstrated that a combination of defactinib and an MDM2 inhibitor produced synergistic or additive effects on cell growth inhibition in a manner determined by the *TP53* status and the inhibitor type.

FAK plays a key role in adhesion of cancer cells to the extracellular matrix, and is consequently related with a metastatic ability and stem cell-like property of the cancer. Mesothelioma often had mutated *NF2* gene and loss of MERLIN functions, and the cells of low-MERLIN expression were sensitive to defactinib since MERLIN-deficient cells were more dependent on FAK for the cell growth than MERLIN-high cells [8]. Nevertheless, the present study demonstrated that the sensitivity to defactinib was not linked with MERLIN expression levels. Kato *et al.* reported that E-cadherin but not MERLIN expression influenced sensitivity to a FAK inhibitor (VS-4718) in mesothelioma [9] and Blum *et al.* demonstrated that the susceptibility of cancer stem cell-enriched mesothelioma cells to defactinib was independent of MERLIN expression [24]. The present study showed that E-cadherin-low NCI-H28 and MSTO-211H cells as demonstrated in the E-cadherin study [9] were not particularly sensitive to defactinib compared with other cells (Fig 2A). The defactinib sensitivity was affected by cell adhesion and the downstream signaling of adhesion molecules, but a number of the molecules contribute to the adhesions and the signal processes. The adhesion is also influenced by the nature of extracellular matrix components *in vivo*. Sensitivity of mesothelioma to the FAK inhibitors is thus subjected to many factors such as the inhibitor types, and is also dependent on experimental systems *in vitro* and on local environmental milieux *in vivo*.

We further examined an effect of defactinib on p53 and MDM2 expression. The current study showed that dephosphorylation of FAK was in general associated with p53 up-regulation or the phosphorylation, but not with MDM2 and the phosphorylation levels. In addition, mutated p53 was known to be resistant to MDM2-mediated degradation [25]. We then presumed that the p53 up-regulation by defactinib was not attributable to the MDM2-mediated effects. The present data

however did not deny a possible reciprocal regulation between FAK and p53 expression levels [18,19]. On the other hand, MDM2 by itself had another function to activate p53 at the transcriptional levels besides ubiquitination of p53, indicating a dual action of MDM2 on p53 expression levels [26]. Furthermore, Kim *et al.* demonstrated MERLIN can facilitate MDM2 degradation and subsequent augmentation of p53 expression [20]. We also found that defactinib-mediated dephosphorylation of FAK was linked with AKT dephosphorylation. Ammoun *et al*. showed that a MERLIN-deleted condition induced phosphorylated AKT and activated MDM2 with subsequent decrease of p53 [27]. This study however demonstrated that the AKT dephosphorylation was not influenced by MERLIN expression but suggested that he dephosphorylation was rather attributable to defactinib-induced growth inhibition since AKT phosphorylation was an active marker for cell growth [28]. These data collectively indicated that defactinib-mediated growth suppression was irrelevant to MERLIN expression, but closely associated with AKT dephosphorylation and generally with p53 augmentation. A previous study indicated that FAK inhibition induced DNA damage and augmented sensitivity to irradiation [29]. We did not precisely examine the DNA damage induction by defactinib since our experimental design was different from the previous study. We however showed that a FAK inhibitor increased susceptibility to a p53-activating DNA damage agent [21] and presume that induction of DNA damage by a FAK inhibitor is the next issue to be investigated.

A combinatory treatment of defactinib and nutlin-3a produced synergistic or additive inhibitory effects on growth of wild-type but not mutated *TP53* cells. Since nutlin-3a did not produce DNA damage response, the nutrin-3a-induced p53 up-regulation and subsequent cell growth inhibition was probably due to inhibition of the p53-MDM2 binding. The nutlin-3a-mediated FAK dephosphorylation was not always linked with p53 phosphorylation or with *TP53* genotype. These data again showed that reciprocal inhibition between p53 and FAK dephosphorylation was dependent on cell types irrespective of the *TP53* genotype. RITA achieved growth inhibitory effects on mesothelioma regardless of *TP53* genotype and a mechanism of the inhibition was different from that of nutlin-3a. Both agents were developed as an inhibitor for p53 and MDM2 interactions, but RITA additionally induced DNA damage responses. Several studies showed that RITA produced cytotoxic effects even in mutated *TP53* cells and the susceptibility to RITA was linked with DNA damage [30,31]. Furthermore, the present study suggested that down-regulated FAK phosphorylation irrespective of *TP53* genotype was attributable to the damage. A previous study also showed that inhibition of FAK activity induced DNA damage responses [29] but it remains unknown whether the RITA-induced DNA damage directly influenced FAK activity. Our study implied that RITA-induced DNA damage was associated with a reciprocal linkage between p53 phosphorylation and FAK dephosphorylation rather than the inhibition of the p53-MDM2 interaction. Chk2 can be involved in RITA-mediated DNA damage since Chk2-defective cells were resistant to RITA- but not nutlin-3a-mediated growth inhibition [32]. We speculate that the ATM-Chk2 pathway contributes to RITA-mediated activation of p53 and subsequent growth suppression. Our previous data showed that a DNA-damaging agent achieved combinatory cytotoxicity with defactinib in wild-type *TP53* mesothelioma cells through down-regulated FAK phosphorylation and p53 up-regulation [21]. Ou *et al.* also reported that combination of nutlin-3a and shRNA for FAK produced additive combinatory effects on growth inhibition of wild-type *TP53* mesothelioma cells through activation of p53 pathways [33]. The present results were consistent with those previous studies and supported the reciprocal regulation between FAK and p53.

Mesothelioma has histologically 3 types, epithelioid, sarcomatoid, and biphasic, and these subtypes have different *TP53* genetic alterations. Epithelioid subtype tends to retain wild-type genotype, whereas sarcomatoid and biphasic types are more likely to harbor the mutations. The present study therefore indicated that the combination of nutlin-3a and defactinib was effective for the epithelioid subtype, while the combination of RITA and defactinib was more suitable for sarcomatoid and biphasic types. Moreover, the sensitivity to defactinib was not linked with the histological types. On the other hand, recent studies have demonstrated that immune checkpoint inhibitors and antiangiogenic agents achieved good clinical effects [34,35]. Expression levels of PD-L1 and VEGF (vascular endothelial growth factor) in mesothelioma *in vivo* can be one of the markers to predict the therapeutic efficacy. Regulation of PD-L1 or VEGF expression through the p53 and FAK signaling pathways also suggests that a possible combination of FAK/MDM2 inhibitors with an immune checkpoint

inhibitor or an antiangiogenic agent may represent next directions to pursue [36,37]. In addition, these small molecule agents need to show high affinity binding to the targets, extend the half-life and enhance the tissue distribution in order to minimize off-target risks and to improve clinical feasibilities. Recent pharmaceutical and clinical developments may enable such nanomolar-level potency in FAK and MDM2 inhibitors [38–40] and facilitate personalized medicine based on patient stratification.

The present study is a preclinical proof-of-concept to explore the synergy between p53 stabilization and FAK inhibition. We showed that a combination of defactinib and a MDM2 inhibitor achieved synergistic or additive inhibitory effects on mesothelioma cell growth, and a potential contribution of FAK and p53 signal pathways to the suppressive effects was associated with selection of MDM2 inhibitors and the *TP53* genotypes (S1B Fig).

## Supporting information

**S1 Fig.** (A) Defactinib-mediated inhibition of mesothelioma cell proliferation with a dye exclusion assay. Cells were seeded in 6 well-plates and treated with defactinib for 24–72 hrs. They were then stained with 0.4% trypan blue solution (Sigma-Aldrich, St. Louis, MO) for 3 minutes at room temperature. The number of stained and unstained cells was counted and the assay were tested in triplicate. (B) A schema of the present study. Mesothelioma cells often have deletion of INK4A/ARF region and MERLIN inactivation. These characteristics induce up-regulated MDM2 expression with subsequent p53 down-regulation. An MDM2 and a FAK inhibitor can augment p53 expression and induce growth inhibition. The current study indicated that nutlin-3a and RITA, a representative MDM2 inhibitor, up-regulated p53 expression via a different mechanism and that the growth inhibition by defactinib, a FAK inhibitor, was unrelated MERLIN expression. A combination of the MDM2 inhibitor and the FAK inhibitor achieved synergistic or additive inhibitory effects, and the effects were linked with the AKT signaling. Nevertheless, the growth inhibitory activity was also subjected to the kind of the MDM2 inhibitor and cells used.
(PDF)

**S2 Fig. (Figure 1 all).** Original blots which were used for Fig 1. Arrows indicate the target molecules. The name of cells was shown in abbreviations. H28: NCI-H28, 211H: MSTO-211H, H2052: NCI-H2052, H226: NCI-H226, H2452: NCI-H2452. EH-10: EHMES-10, EH-1: EHMES-1.
(PDF)

**S3 Fig. (Figure 2B FAK).** Original blots which were used for Fig 2B FAK expression. Arrows indicate the target molecules. The name of cells was shown in the abbreviations. We did not use a photo of 8 µM defactinib treatments. In some of the blots, we used the same blot to detect others molecules without stripping the blot and consequently showed the target molecules by the arrows.
(PPTX)

**S4 Fig. (Figure 2B p-FAK).** Original blots which were used for Fig 2B phosphorylated FAK expression. Arrows indicate the target molecules. The name of cells was shown in the abbreviations. We did not use a photo of 8 µM defactinib treatments. In some of the blots, we used the same blot to detect others molecules without stripping the blot and consequently showed the target molecules by the arrows.
(PPTX)

**S5 Fig. (Figure 2B p53).** Original blots which were used for Fig 2B p53 expression. Arrows indicate the target molecules (NCI-H2452 had a truncated p53). The name of cells was shown in the abbreviations.
(PPTX)

**S6 Fig. (Figure 2B p-p53).** Original blots which were used for Fig 2B phosphorylated p53 expression. Arrows indicate the target molecules (NCI-H2452 had a truncated p53). The name of cells was shown in the abbreviations. We did not use a

photo of 8 μM defactinib treatments. In some of the blots, we used the same blot to detect others molecules without stripping the blot and consequently showed the target molecules by the arrows.
(PPTX)

**S7 Fig. (Figure 2B AKT).** Original blots which were used for Fig 2B AKT expression. Arrows indicate the target molecules. The name of cells was shown in the abbreviations. We did not use a photo of 8 μM defactinib treatments.
(PPTX)

**S8 Fig. (Figure 2B p-AKT).** Original blots which were used for Fig 2B phosphorylated AKT expression. Arrows indicate the target molecules. The name of cells was shown in the abbreviations. We did not use a photo of 8 μM defactinib treatments. In some of the blots, we used the same blot to detect others molecules without stripping the blot and consequently showed the target molecules by the arrows.
(PPTX)

**S9 Fig. (Figure 2B MDM2).** Original blots which were used for Fig 2B MDM2 expression. Arrows indicate the target molecules (both 90 kDa and 60 kDa molecules). The name of cells was shown in the abbreviations.
(PPTX)

**S10 Fig. (Figure 2B p-MDM2).** Original blots which were used for Fig 2B phosphorylated MDM2 expression. Arrows indicate the target molecules (90 kDa). The name of cells was shown in the abbreviations. We did not use a photo of 8 μM defactinib treatments.
(PPTX)

**S11 Fig. (Figure 2B Caspase-9).** Original blots which were used for Fig 2B caspase-9 and the cleaved caspase-9 expressions. The antibody detected the cleaved form. Arrows indicate the target molecules (both original and cleaved molecules). The name of cells was shown in the abbreviations. We did not use a photo of 8 μM defactinib treatments. In some of the blots, we used the same blot to detect others molecules without stripping the blot and consequently showed the target molecules by the arrows.
(PPTX)

**S12 Fig. (Figure 2B PARP).** Original blots which were used for Fig 2B PARP and the cleaved PARP expressions. The antibody detected the cleaved form. Arrows indicate the target molecules (both original and cleaved molecules). The name of cells was shown in the abbreviations. We did not use a photo of 8 μM defactinib treatments. In some of the blots, we used the same blot to detect others molecules without stripping the blot and consequently showed the target molecules by the arrows.
(PPTX)

**S13 Fig. (Figure 2B Tubulin).** Original blots which were used for Fig 2B tubulin expression. Arrows indicate the target molecules. The name of cells was shown in the abbreviations. We did not use a photo of 8 μM defactinib treatments.
(PPTX)

**S14 Fig. (Figure 3B p53 and p-p53).** Original blots which were used for Fig 3B p53 and phosphorylated p53 expressions. Arrows indicate the target molecules. The name of cells was shown in the abbreviations. In some of the blots, we used the same blot to detect others molecules without stripping the blot and consequently showed the target molecules by the arrows.
(PPTX)

**S15 Fig. (Figure 3B FAK and p-FAK).** Original blots which were used for Fig 3B FAK and phosphorylated FAK expressions. Arrows indicate the target molecules. The name of cells was shown in the abbreviations. In some of the blots, we

used the same blot to detect others molecules without stripping the blot and consequently showed the target molecules by the arrows.
(PPTX)

**S16 Fig. (Figure 3B p-H2AX).** Original blots which were used for Fig 3B p-H2AX expression. Arrows indicate the target molecules. The name of cells was shown in the abbreviations. We did not use a photo of 15 µM nutlin-3a treatments. In some of the blots, we used the same blot to detect others molecules without stripping the blot and consequently showed the target molecules by the arrows.
(PPTX)

**S17 Fig. (Figure 3B Actin).** Original blots which were used for Fig 3B actin expression. Arrows indicate the target molecules. The name of cells was shown in the abbreviations. We did not use a photo of 15 µM nutlin-3a treatments.
(PPTX)

**S18 Fig. (Figure 4B p53 and p-p53).** Original blots which were used for Fig 4B p53 and phosphorylated p53 expressions. Arrows indicate the target molecules. The name of cells was shown in the abbreviations. In some of the blots, we used the same blot to detect others molecules without stripping the blot and consequently showed the target molecules by the arrows.
(PPTX)

**S19 Fig. (Figure 4B FAK and p-FAK).** Original blots which were used for Fig 4B FAK and phosphorylated FAK expressions. Arrows indicate the target molecules. The name of cells was shown in the abbreviations. In some of the blots, we used the same blot to detect others molecules without stripping the blot and consequently showed the target molecules by the arrows.
(PPTX)

**S20 Fig. (Figure 4B p-H2AX and Actin).** Original blots which were used for Fig 4B p-H2AX and actin expressions. Arrows indicate the target molecules. The name of cells was shown in the abbreviations. We did not use a photo of 6 µM RITA treatments. In some of the blots, we used the same blot to detect others molecules without stripping the blot and consequently showed the target molecules by the arrows.
(PPTX)

**S21 Fig. (Figure 7 FAK).** Original blots which were used for Fig 7 FAK expression. Arrows indicate the target molecules. The name of cells was shown in the abbreviations. Abbreviations. NT: no treatment shown as (-) in Fig 7, Def: defactinib, Nut: nutlin-3a, N+D: nutlin-3a+defactinib, R+D: RITA+defactinib.
(PPTX)

**S22 Fig. (Figure 7 p-FAK).** Original blots which were used for Fig 7 phosphorylated FAK expression. Arrows indicate the target molecules. The name of cells was shown in the abbreviations. Abbreviations used for treatment were shown in S21 Fig.
(PPTX)

**S23 Fig. (Figure 7 p53).** Original blots which were used for Fig 7 p53 expression. Arrows indicate the target molecules. The name of cells was shown in the abbreviations. Abbreviations used for treatment were shown in S21 Fig.
(PPTX)

**S24 Fig. (Figure 7 p-p53).** Original blots which were used for Fig 7 phosphorylated p53 expression. Arrows indicate the target molecules. The name of cells was shown in the abbreviations. Abbreviations used for treatment were shown in S21 Fig. In some of the blots, we used the same blot to detect others molecules without stripping the blot and consequently showed the target molecules by the arrows.
(PPTX)

**S25 Fig. (Figure 7 AKT and p-AKT).** Original blots which were used for Fig 7 AKT and phosphorylated AKT expressions. Arrows indicate the target molecules. The name of cells was shown in the abbreviations. Abbreviations used for treatment were shown in S21 Fig. In some of the blots, we used the same blot to detect others molecules without stripping the blot and consequently showed the target molecules by the arrows.
(PPTX)

**S26 Fig. (Figure 7 PARP).** Original blots which were used for Fig 7 PARP and cleaved PARP expressions. The antibody detected the cleaved form. Arrows indicate the target molecules (both original and cleaved molecules). The name of cells was shown in the abbreviations. Abbreviations used for treatment were shown in S21 Fig. In some of the blots, we used the same blot to detect others molecules without stripping the blot and consequently showed the target molecules by the arrows.
(PPTX)

**S27 Fig. (Figure 7 Actin).** Original blots which were used for Fig 7 actin expression. Arrows indicate the target molecules. The name of cells was shown in the abbreviations. Abbreviations used for treatment were shown in S21 Fig.
(PPTX)

**S1 Table. Information of cells used in the study.**
(DOCX)

**S2 Table. Information of antibody used in the study.** Dilution of primary antibody and information of secondary antibody used in the study.
(DOCX)

**S3 Table. Molecular expression levels in Fig 1.** Expression of the molecules in Fig 1 was quantified with ImageJ software (NIH, Bethesda, MD, USA). The intensity of target protein bands was normalized to the intensity of tubulin as a loading control. Respective protein expression levels of NCI-H28 were used as a standard (expressed as 1.00).
(DOCX)

**S4 Table. Molecular expression levels in Fig 2B.** Expression of the molecules in Fig 2B was quantified with ImageJ software (NIH, Bethesda, MD, USA). The intensity of target protein bands was normalized to the intensity of tubulin as a loading control. Respective protein expression levels of untreated cells were used as a standard (expressed as 1.00).
(DOCX)

**S5 Table. Molecular expression levels in Fig 3B.** Expression of the molecules in Fig 3B was quantified with ImageJ software (NIH, Bethesda, MD, USA). The intensity of target protein bands was normalized to the intensity of actin as a loading control. Respective protein expression levels of untreated cells were used as a standard (expressed as 1.00).
(DOCX)

**S6 Table. Molecular expression levels in Fig 4B.** Expression of the molecules in Fig 4B was quantified with ImageJ software (NIH, Bethesda, MD, USA). The intensity of target protein bands was normalized to the intensity of actin as a loading control. Respective protein expression levels of untreated cells were used as a standard (expressed as 1.00).
(DOCX)

**S7 Table. Molecular expression levels in Fig 7.** Expression of the molecules in Fig 7 was quantified with ImageJ software (NIH, Bethesda, MD, USA). The intensity of target protein bands was normalized to the intensity of actin as a loading control. Respective protein expression levels of untreated cells were used as a standard (expressed as 1.00).
(DOCX)

## Author contributions

**Conceptualization:** Masatoshi Tagawa, Naoto Yamaguchi.

**Data curation:** Thảo Thi Thanh Nguyễn, Yuji Tada, Hideaki Shimada, Kenzo Hiroshima.

**Funding acquisition:** Masatoshi Tagawa.

**Investigation:** Xuerao Ning, Thảo Thi Thanh Nguyễn, Takao Morinaga.

**Methodology:** Xuerao Ning, Thảo Thi Thanh Nguyễn, Takao Morinaga, Kenzo Hiroshima.

**Project administration:** Masatoshi Tagawa, Naoto Yamaguchi.

**Resources:** Xuerao Ning, Takao Morinaga.

**Software:** Takao Morinaga, Yuji Tada, Hideaki Shimada, Kenzo Hiroshima.

**Supervision:** Masatoshi Tagawa, Naoto Yamaguchi.

**Validation:** Yuji Tada, Hideaki Shimada, Kenzo Hiroshima.

**Visualization:** Yuji Tada.

**Writing – original draft:** Masatoshi Tagawa, Thảo Thi Thanh Nguyễn, Naoto Yamaguchi.

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
