## [Decision Letter · Decision Letter 0]

26 Sep 2025

Dear Dr. Tagawa,

Thank you for submitting your manuscript to PLOS ONE. After careful consideration, we feel that it has merit but does not fully meet PLOS ONE’s publication criteria as it currently stands. Therefore, we invite you to submit a revised version of the manuscript that addresses the points raised during the review process.

Please find the detailed comments from the reviewers as below. We kindly ask you to carefully address each point raised in your revision. When submitting the revised manuscript, please also provide a point-by-point response to the reviewers’ comments, outlining the changes made or explaining your reasoning if any suggestions were not incorporated.

We look forward to receiving your revised manuscript.

Kind regards,

Zu Ye, Ph.D.

Academic Editor

PLOS ONE

Journal Requirements:

Reviewers' comments:

Reviewer's Responses to Questions

**Comments to the Author**

1. Is the manuscript technically sound, and do the data support the conclusions?

Reviewer #1: Yes

Reviewer #2: Yes

2. Has the statistical analysis been performed appropriately and rigorously?

Reviewer #1: Yes

Reviewer #2: No

3. Have the authors made all data underlying the findings in their manuscript fully available?

Reviewer #1: Yes

Reviewer #2: Yes

4. Is the manuscript presented in an intelligible fashion and written in standard English?

Reviewer #1: Yes

Reviewer #2: Yes

Reviewer #1: The authors investigated the combination effect of FAK and MDM2 inhibitors using mesothelioma cell lines. They showed that FAK inhibitor defactinib inhibited cell growth and induced FAK dephosphorylation irrespective of the TP53 genotype, and that the inhibited FAK phosphorylation was not associated with MERLIN levels or with p53 up-regulation but linked with AKT dephosphorylation. This article may include some interesting findings. However, the following issues should be addressed.

#1. NCI-H226 is used as a mesothelioma cell line. The website of ATCC shows that this line exhibits epithelial morphology that was isolated from the lungs of a male with squamous cell carcinoma. The website of cBioPortal also shows that the type of this line is lung squamous cell carcinoma. The authors should confirm that what type of cell line NCI-H226 is.

#2. The authors concluded that a combination of defactinib and the MDM2 inhibitors in general produced synergistic or additive effects to inhibit growth of mesothelioma. However, as they describe, these effects were different depending on MDM2 inhibitors (nultin-3a or RITA) and cell lines tested. This study lacks impact.

#3. The part of “Discussion” is redundant.

Reviewer #2: Mesothelioma has characteristic genetic changes including inactivation of neurofibromatosis type 2 (NF2) and deletion of the INK4A/ARF region. This study analyzed the combinatory effects of the FAK and MDM2 inhibitors and that consequent signaling pathway changes. The results showed that combination of FAK inhibitor defactinib and the MDM2 inhibitors in general produced synergistic or additive effects to inhibit growth of mesothelioma, but these effects were influenced by MDM2 inhibitors and cell lines tested. However, there are some concerns as follows:

1. The FAK inhibitor defactinib and the MDM2 inhibitor nutlin-3a were at the micromolar scale, however, the small molecular inhibitor drugs are usually at 10-20 nanomole scale. How about the future drug development?

2. The combination effects were mutation and cell line specific. How to combine the findings of this study with the subgroup mesothelioma patients?

3. The immune check point inhibitors have made great success in the treatment of mesothelioma patients. How about the influence of both FAK and MDM2 inhibitors to the expression level of PD-L1 in mesothelioma cells?

4. The antiangiogenic therapy has been proved valuable in relieving pleural effusion. How about the influence of both FAK and MDM2 inhibitors to the expression level of VEGF in mesothelioma cells?

5. There is no proper statistics in the figures and figure legends. All the WB results should have statistics of gray values.

**Do you want your identity to be public for this peer review?** For information about this choice, including consent withdrawal, please see our Privacy Policy

Reviewer #1: No

Reviewer #2: No

---

## [Author Response · Author response to Decision Letter 1]

12 Nov 2025

Replies to Reviewers

(A) Comments from Reviewer #1

(1) NCI-H226 is used as a mesothelioma cell line. The website of ATCC shows that this line exhibits epithelial morphology that was isolated from the lungs of a male with squamous cell carcinoma. The website of cBioPortal also shows that the type of this line is lung squamous cell carcinoma. The authors should confirm that what type of cell line NCI-H226 is.

[Reply]

Thank you very much for your comment. The ATCC website also showed that this cell line was mesothelioma in addition to lung carcinoma. As mentioned by the reviewer, cBioPortal showed that the cell line was classified as lung carcinoma. The Biohippo and the Cellosaurus websites however mentioned that NCI-H226 was mesothelioma cell line (pleural epithelioid mesothelioma). The confusion or the ambiguity came from several reasons. For example, the cells were derived from pleural effusion of a lung cancer patient and histology of the two cancer types was overlapped. Nevertheless, genetic markers supported that NCI-H226 cells were rather mesothelioma. NCI-H226 cells showed wild-type TP53 genotype, deletion of the INK4A/ARF region and mutated NF2 gene (according to Cancer Cell Line Encyclopedia, Catalogue of Somatic Mutations in Cancer, and Cancer Dependency Map). Moreover, NCI-H226 cells strongly positive for mesothelin which was a characteristic protein marker of mesothelium and mesothelioma, but non-small cell lung cancer was negative for mesothelin. While some studies classified the NCI-H226 cell line as lung carcinoma based on sources like ATCC, many others used it as a model for mesothelioma. Given its genetic profile and mesothelin expression, we consider it more appropriate to regard NCI-H226 as a mesothelioma cell line, in line with the majority of previous publications.

(2) The authors concluded that a combination of defactinib and the MDM2 inhibitors in general produced synergistic or additive effects to inhibit growth of mesothelioma. However, as they describe, these effects were different depending on MDM2 inhibitors (nultin-3a or RITA) and cell lines tested. This study lacks impact.

[Reply]

Thank you for your comment. We understand the concern that the combinatory effects varied and were to some extent dependent on the type of MDM2 inhibitors and the mesothelioma cell lines used, which may reduce the overall impact of the study. Initially, we considered focusing on nutlin-3a in wild-type p53 cells to simplify our conclusions. However, we expanded the study to include RITA and mutated p53 cells to explore the broader applicability of the combination. While both nutlin-3a and RITA inhibit the MDM2-p53 interaction, RITA also induces DNA damage, which can elevate p53 levels independently. The combinatory effects were therefore influenced by the inhibitors. Our results showed that both MDM2 inhibitor and defactinib produced synergistic or additive inhibitory effects in most of wild-type p53 cells except for NCI-H226 cells treated at a low RITA concentration. In mutated p53 cells treated with nutlin-3a, the combinatory effects were antagonistic. In the case of RITA-treated cells, the combination produced synergistic effects in irrespective of the TP53 genotype except NCI-226 cells as mention above and H2452 cells with truncated p53.

Furthermore, the effects of MDM2 inhibitors on mutated p53 is influenced by the p53 structure and the MDM2-mediated effects could be altered in mutated p53 cells. Even if the MDM2 inhibitors increases expression levels of mutated p53, the up-regulation would rather support for cell survival but not growth inhibition. The combinatory effects in mutated p53 cells could be therefore different from those in wild-type p53 cells. Nevertheless, the combination of RITA and defactinib achieved synergistic effects in mutated p53 cells. On the other hand, NCI-H2452 cells with truncated p53 and EHMES-10 cells with resistance to nutrin-3a and RITA showed different responses from wild-type p53 cells. The extended experiments thus revealed that the combinatory effects were influenced by MDM2 inhibitors and cells used although the combination produced synergistic or additive effects in most of the cases. We have revised the manuscript to clarify our conclusions and better reflect the scope and limitations of the study. Please see the revised part on page 4-5 line 67-71, on page 8 line 110-111 in unmarked manuscript, and Discussion in marked manuscript.

(3) The part of “Discussion” is redundant.

[Reply]

Thank you very much for your comment. We have revised the Discussion section to enhance clarity and eliminate redundancy. Please refer to the revised parts on page 22-27 in marked manuscript.

(B) Comments from Reviewer #2

(1) The FAK inhibitor defactinib and the MDM2 inhibitor nutlin-3a were at the micromolar scale, however, the small molecular inhibitor drugs are usually at 10-20 nanomole scale. How about the future drug development?

[Reply]

Thank you very much for your insightful point. As the reviewer pointed out, small-molecule inhibitor drugs used in clinical settings typically exhibit efficacy at nanomolar concentrations, whereas the compounds used in our study were applied at micromolar levels. The present study is preclinical and intended to demonstrate proof-of-concept to explore a strategy for up-regulating endogenous wild-type p53 levels. While some molecular-targeted agents are effective at micromolar concentrations, such doses may increase off-target risks in clinical applications. Future drug development should therefore aim for high target affinity binding, extended half-life and enhanced tissue distribution to improve clinical feasibilities. We added a statement regarding this possible future direction in Discussion of the revised manuscript (See on page 27 line 433-436 with references, No 38 and 39 in unmarked manuscript).

(2) The combination effects were mutation and cell line specific. How to combine the findings of this study with the subgroup mesothelioma patients?

[Reply]

Thank you very much for your valuable comment. The reviewer referred to the histological subtypes of mesothelioma-epithelioid, sarcomatoid, and biphasic-as well as their associated genetic alterations. In general, the epithelioid subtype tends to retain wild-type TP53 genotype, whereas sarcomatoid and biphasic types are more likely to harbor TP53 mutations; consequently, the epithelioid subtype is more susceptible to combination therapy aimed at up-regulating wild-type p53. On the other hand, RITA increases susceptibility to the combination even in cells with mutated p53. Based on our findings, we suggest that the combination of nutlin-3a and defactinib is effective for the epithelioid subtype, while the combination of RITA and defactinib is more suitable for sarcomatoid and biphasic types. We discussed these possible relationships between histological subtypes and combination effects in the revised manuscript (page 26-27 line 422-427 in unmarked manuscript). In addition, sensitivity to defactinib was not linked to TP53 mutation status (page 27 line 427 in unmarked manuscript).

(3) The immune check point inhibitors have made great success in the treatment of mesothelioma patients. How about the influence of both FAK and MDM2 inhibitors to the expression level of PD-L1 in mesothelioma cells?

[Reply]

Thank you very much for highlighting the importance of immune checkpoint inhibitors in mesothelioma treatment. We fully agree with the reviewer’s point. Although PD-L1 expression has been widely studied, its predictive value in mesothelioma patients receiving immune checkpoint inhibitors remains inconclusive at this moment as far as we are aware. Nevertheless, FAK inhibition may reduce PD-L1 expression and p53 is known to down-regulates PD-L1 expression. We therefore believe that a potential combination of FAK inhibitors, MDM2 inhibitors and immune check point inhibitors represents an interesting research direction. We however presume that investigating PD-L1 expression following treatment with FAK and MDM2 inhibitors will be difficult to include in this study since it may fall outside the scope of the present study. Such analyses may be the next investigation because it requires separate, dedicated studies. We hope that the future studies, possibly including in vivo models or patient-derived samples, will clarify the impact of these inhibitors on PD-L1 expression and their potential synergy with immune checkpoint therapies. We appreciate the reviewer’s suggestion and have included this point in the revised Discussion section on page 27 line 428-433 with references in unmarked manuscript.

(4) The antiangiogenic therapy has been proved valuable in relieving pleural effusion. How about the influence of both FAK and MDM2 inhibitors to the expression level of VEGF in mesothelioma cells?

[Reply]

Thank you very much for your comment. As the reviewer mentioned, recent studies indicated that inhibition of angiogenesis was effective, particularly when combined with chemotherapy and immunotherapy. Nevertheless, the clinical value of VEGF expression as a predictive marker remains unclear at this time. On the other hand, FAK is involved in VEGF regulation and angiogenesis, and p53 up-regulation can negatively regulate VEGF transcription. These findings suggest that FAK and MDM2 inhibitors can potentially enhance efficacy of the antiangiogenesis therapy. We however believe that analyzing the VEGF expression following the treatments is important but may fall beyond the scope of the current work at this moment. We consider this a promising direction for future research and have mentioned the potential combination with antiangiogenesis therapy in revised Discussion section (page 27 line 428-433 and references in unmarked manuscript).

(5) There is no proper statistics in the figures and figure legends. All the WB results should have statistics of gray values.

[Reply]

We appreciate the reviewer’s comment regarding statistical analysis of the Western blot data. In our study, we performed densitometric quantification using ImageJ and normalized each band to the corresponding loading control. We think at this moment that it is unfortunately not feasible to repeat all the analyses from the original materials due to the large number of samples. We conducted multiple Western blot experiments to obtain clear and consistent data, and selected representative results for presentation in the figures. We therefore are confident in the accuracy and reproducibility of the data. While we acknowledge that statistical testing across biological replicates is ideal, the scope and limitations of our dataset make this approach challenging. We have clarified this point in the revised manuscript and added a note in the figure legend (Figure 1, 2B, 3B, 4B, 7B) to explain the methodology. We hope this explanation addresses the reviewer’s concern and respectfully request understanding given the constraints of our study.

---

## [Decision Letter · Decision Letter 1]

22 Dec 2025

Dear Dr. Tagawa,

Thank you for submitting your manuscript to PLOS ONE. After careful consideration, we feel that it has merit but does not fully meet PLOS ONE’s publication criteria as it currently stands. Therefore, we invite you to submit a revised version of the manuscript that addresses the points raised during the review process.

We look forward to receiving your revised manuscript.

Kind regards,

Zu Ye, Ph.D.

Academic Editor

PLOS One

Journal Requirements:

Reviewers' comments:

Reviewer's Responses to Questions

**Comments to the Author**

Reviewer #1: All comments have been addressed

Reviewer #3: (No Response)

2. Is the manuscript technically sound, and do the data support the conclusions?

Reviewer #1: Yes

Reviewer #3: No

3. Has the statistical analysis been performed appropriately and rigorously?

Reviewer #1: Yes

Reviewer #3: No

4. Have the authors made all data underlying the findings in their manuscript fully available?

Reviewer #1: Yes

Reviewer #3: Yes

5. Is the manuscript presented in an intelligible fashion and written in standard English?

Reviewer #1: Yes

Reviewer #3: Yes

Reviewer #1: The authors revised the manuscript based on the comments by the reviewers and the manuscript became much better.

Reviewer #3: This is a systematic but incremental in vitro study with some remaining significant methodological and analytical concerns. Is not that easy to draw sound valid conclusions from these experimental schemes. On the good side it is using a comprehensive cell line panel with varied TP53, INK4/ARF genotypes. They testing both nutlin-3a and RITA (which also induces DNA damage). Though it is clear that none of these molecules are really in clinical progress and there are more potent MDM2 inhibitors like idasanutlin.

Results are also not really surprising for the basics, like RITA works in mutant p53 cells, induces DNA damage. FAK-p53 is confusing. Synergy mechanism unclear authors admit not linked to expected molecular changes. This study is refinement rather than original novel findings since there is previous work on p53-activating agents, FAK inhibitors and so on.

The immunblots and statistics in paper is an issue. This is not a matter of scope and limitations it's fundamental scientific rigor. The study needs substantial strengthening before publication.

Main concerns

Immunonblotting is not easy methodology, and I think authors did a good job, showing all blots, full scan in supplement. But there are major issues with MDM2, gH2AX, dirty antibodies and comparison issues. Control for antibodies. Also, authors state they conducted multiple immunoblot experiments but show only representative results. Response to reviewer 2 is to some extent inadequate "it is unfortunately not feasible to repeat all the analyses". They claim confidence in accuracy and reproducibility without providing statistics. This becomes more of an issue given concerns below. In general a better approach is statistical analysis of western blots from multiple independent experiments, time and dose response. Add blot statistics from ≥3 independent experiments. Quantify blots with error bars and statistical tests where effect is limited or vague/unclear. Clarify blotting methodology approach and add time course and dose experiments (not just 24 hrs).

I understand the issue of repeating many experiments and that some are repeated within the paper from figure to figure. However, regardless of this there are major issues remaining.

1.There is a major issue with MDM2 (figure 2), as it is shown in main figure, it looks like MDM2 and p-MDM2 bands are similar in size and overlap, but they are not looking at the full scan the bands don’t fully match. It is not possible to deduce what we are looking at or that the MDM2 bands are what they are. Also no size markers for the main figure. In supplement is clear non-modified MDM2 bands do not correspond to sizes of p-MDM2. And, it is not clear that this is actually MDM2 to begin with. What is the evidence that p90, p60 bands as shown is a actually MDM2?

2.Second major issue concerns comparability of basal gH2AX (figure 3 versus figure 4, respective cell line untreated condition). The level of gH2AX in untreated cells (respective cells) varies a lot between experiments for the same cell line. Presumably this could be to different exposure times,between lines/experiments though this does not seem to be an issue for p53. This makes it very difficult to judge and compare gH2AX. Besides, the antibody is dirty, many bands.

3.Third major issue, how come there is such increase in p-p53 upon Nutlin treatment same almost as with RITA? It looks quite a lot, though corresponding to total levels. Figure 3 vs 4. This nutlin effect on p53-p is not really that apparent in figure 7 e g in NCI H28, this really contracts within the paper. Similar to gH2AX.

4.Drugs and cell lines. IC50 values in 1-6 micromolar range is not so good actually, effects vary dramatically between cell lines (though expected). EHMES-10 resistant to both MDM2 inhibitors it looks like and NCI-H226 shows unusual responses in my opinion. Variability limits generalizability and clinical translation.

5.Direct comparisons difficult since only one exposure duration tested.

6.Figure 7: Complex dosing scheme makes interpretation difficult

7.Discussion is still somewhat redundant despite revisions

8.Abstract conclusion is very vague just says "effects depend on genotype and inhibitor type"

9. The title is not really what the data shows, it is a bit confusing.

**Do you want your identity to be public for this peer review?** For information about this choice, including consent withdrawal, please see our Privacy Policy

Reviewer #1: No

Reviewer #3: No

---

## [Author Response · Author response to Decision Letter 2]

3 Feb 2026

Replies to Reviewer #3

Major concerns

Immunonblotting is not easy methodology, and I think authors did a good job, showing all blots, full scan in supplement. But there are major issues with MDM2, gH2AX, dirty antibodies and comparison issues. Control for antibodies. Also, authors state they conducted multiple immunoblot experiments but show only representative results. Response to reviewer 2 is to some extent inadequate "it is unfortunately not feasible to repeat all the analyses". They claim confidence in accuracy and reproducibility without providing statistics. This becomes more of an issue given concerns below. In general a better approach is statistical analysis of western blots from multiple independent experiments, time and dose response. Add blot statistics from ≥3 independent experiments. Quantify blots with error bars and statistical tests where effect is limited or vague/unclear. Clarify blotting methodology approach and add time course and dose experiments (not just 24 hrs). I understand the issue of repeating many experiments and that some are repeated within the paper from figure to figure. However, regardless of this there are major issues remaining.

[Reply]

We appreciate the reviewer’s constructive comments regarding our immunoblotting methodology. We have addressed the specific concerns regarding MDM2 and γH2AX (p-H2AX) in the subsequent itemized responses as below. Here, we clarify the rationale for our choice of treatment duration and dosage.

For defactinib, a concentration of 8 µM resulted in excessive cell death, which interfered with the detection of clear FAK bands. Furthermore, while the treatment beyond 24 hrs effectively induced FAK dephosphorylation, prolonged exposure led to significant cytotoxicity. Consequently, we selected a concentration range of 1–4 µM and a 24-hrs incubation period to demonstrate FAK inhibition without compromising cell viability.

Regarding nutlin-3a and RITA, the concentrations used in Figures 3B (nutlin-3a; 0, 5 and 10 microM) and 4B (RITA; 0, 1.5, 3 microM) were optimized to ensure p53 induction without significant toxicity. Since p53 expression levels remained consistent for 24 hrs after the defactinib treatment and across 24, 48, and 72 hrs of the nutlin-3a/RITA treatment, we standardized the incubation period to 24 hrs. This duration was also applied to the combination treatments to maximize p53 stability while minimizing defactinib-induced cell death, thereby ensuring the reliability of the immunoblotting results. We therefore conducted preliminary experiments and selected the optimal condition regarding treatment time and dose.

Comment (1) 1 There is a major issue with MDM2 (figure 2), as it is shown in main figure, it looks like MDM2 and p-MDM2 bands are similar in size and overlap, but they are not looking at the full scan the bands don’t fully match. It is not possible to deduce what we are looking at or that the MDM2 bands are what they are. Also no size markers for the main figure. In supplement is clear non-modified MDM2 bands do not correspond to sizes of p-MDM2. And, it is not clear that this is actually MDM2 to begin with. What is the evidence that p90, p60 bands as shown is a actually MDM2?

[Reply]

Thank you for pointing out the inconsistencies regarding the MDM2 and p-MDM2 blots. We realized our mistakes of p-MDM blots in Figure 2B and Supplementary Figure 10 (the corresponding original full blot) and noticed labeling errors in both of the figures.

Specifically, the bands previously labeled as 90 kDa and 60 kDa in Supplementary Figure 10 were incorrect. Upon re-examination of molecular weight markers and additional sample blots (see below), the band previously indicated as 90 kDa was a non-specific signal appearing above 100 kDa, while the band previously labeled as 60 kDa is the authentic p-MDM2 at 90 kDa. Authentic 60 kDa p-MDM2 bands were quite low in the intensity with almost background levels because 60 kDa MDM-2 molecules lost the phosphorylation site at Ser 166 and thereby the anti-p-MDM2 antibody used in the study could not detect 60 kDa p-MDM2. We then removed the 60 kDa p-MDM2 arrows in the revised Figure 2B and Supplementary Figure 10.

We have up-loaded new Figure 1, 2, Supplementary Figure 10 and Supplementary Table 4 to address your concerns.

(a) new Figure 2B & Supplementary Figure 10: We have replaced the p-MDM2 blots with the correct 90 kDa bands and added molecular weight markers as requested. We have also revised Supplementary Figure 10 legend (line 737 in page 45). The identification of the 90 kDa band as MDM2/p-MDM2 is based on the manufacturer's datasheet (Santa Cruz Biotechnology and Cell Signaling) and its alignment with the protein standards.

(b) new Supplementary Table 4: We have updated the data. We removed the non-specific (previously "90 kDa") data and provided the intensity data for the authentic 90 kDa p-MDM2 (previously "60 kDa"). Despite these labeling corrections, the trend of p-MDM2 levels in response to defactinib treatment remained consistent with our original findings; the intensity data of non-specific (previously “90 kDa”) bands were similar to those of authentic 90 kDa p-MDM2 (previously "60 kDa"). The overall conclusion of the study thereby remains unchanged.

(c) Size markers: We added the size markers in MDM2 and p-MDM2 blots of revised Figure 2B and also in Figure 1 to ensure clarity.

We apologize for the confusion caused by the initial mislabeling.

Comment (2) 2 Second major issue concerns comparability of basal gH2AX (figure 3 versus figure 4, respective cell line untreated condition). The level of gH2AX in untreated cells (respective cells) varies a lot between experiments for the same cell line. Presumably this could be to different exposure times,between lines/experiments though this does not seem to be an issue for p53. This makes it very difficult to judge and compare gH2AX. Besides, the antibody is dirty, many bands.

[Reply A]

Thank you for your comment. The reviewer is correct that the level of p-H2AX in untreated cells appeared different between Figure 3B and 4B even for the same cell line. This discrepancy is due to different exposure times, which were optimized to capture the specific effects of each inhibitor. We needed to show whether nutlin-3a or RITA induced DNA damage, and consequently we had to use a different exposure time depending on the property of the inhibitor.

(a) Figure 3B (nutlin-3a): All the cells showed relatively high expression levels even in untreated cells. Since nutlin-3a did not induce DNA damage, the p-H2AX signals were inherently weak across all the samples. We intentionally exposed the filters for a long period until the basic signals in untreated cells became clearly detectable in order to confirm whether the western blotting was technically successful and to demonstrate the lack of p-H2AX induction, A short exposure showed no p-H2AX signal in all the lanes, which made it difficult to judge whether the western blots were properly conducted.

(b) Figure 4B (RITA): In contrast, RITA is a potent inducer of DNA damage. We used a shorter exposure time to prevent the strong signals in RITA-treated cells from reaching saturation. Consequently, the basal levels in untreated cells (with the exception of MSTO-211H) appeared undetectable under these conditions. MSTO-211H cells showed less sensitivity to RITA-induced DNA damage compared with other cells. To visualize their response, we applied a longer exposure specifically to this cell line. We therefore added sentences in Figure 3B and 4B legends as follows to explain more in detail. We also revised Results section to explain results of RITA-treated MSTO-211H cells.

(a) Figure 3 legend (lines 640-642 in page 40)

For phosphorylated H2AX, the blots were exposed for a longer period to ensure the detection of basal signals in untreated cells. The basal intensity therefore differed from that of the RITA experiments in Figure 4B where shorter exposures were used.

(b) Figure 4 Legend (lines 651-652 in page 40)

For phosphorylated H2AX of MSTO-211H cells, the blot was exposed for a longer period to ensure the detection of basal signals in untreated cells.

(c) Results (line 274-275 in page 17)

(Before the revision)

All the RITA-treated cells showed decreased FAK phosphorylation levels and increased phosphorylation of H2AX.

(After the revision)

All the RITA-treated cells showed decreased FAK phosphorylation and increased H2AX phosphorylation levels, with the exception of MSTO-211H cells exhibiting a minor H2AX increase.

[Reply B]

Regarding the "dirty" appearance and multiple bands mentioned by the reviewer, this was also a result of the extended exposure times required for the nutlin-3a experiments (as seen in Supplementary Figure 16, which is for Figure 3B p-H2AX). The high background and non-specific bands emerged only under prolonged exposure to visualize the weak basal signals. For nutlin-3a treatment, we needed to expose the filters for a long period to obtain the signal in untreated cells since nutlin-3a did not induce DNA damage. In contrast, the blots in Supplementary Figure 20 (RITA treatment, used for Figure 4B p-H2AX) showed a cleaner background due to the shorter exposure time which enabled us to detect the DNA damage signal. We used the same antibody lot from BioLegend for all experiments, and we would like to maintain that the extra bands were an artifact of overexposure rather than a lack of antibody specificity.

Comment (3) 3.Third major issue, how come there is such increase in p-p53 upon Nutlin treatment same almost as with RITA? It looks quite a lot, though corresponding to total levels. Figure 3 vs 4. This nutlin effect on p53-p is not really that apparent in figure 7 e g in NCI H28, this really contracts within the paper. Similar to gH2AX.

[Reply]

Thank you for this critical observation regarding the p-p53 levels. We understand the reviewer's concern about the apparent discrepancy between Figure 3B and Figure 7. The reviewer mentioned that p-p53 in the wild-type TP53 cells treated with nutlin-3a significantly increased as much as that in the cells treated with RITA, and questioned that the nutlin-3a-mediated up-regulation of p-p53 was not significant in Figure 7. The difference in p-p53 induction levels between these figures is primarily due to the different concentrations of nutlin-3a used in each experiment, as detailed below.

(a) Concentrations: In Figure 3B, we aimed to demonstrate the maximum biological effect of nutlin-3a as a single agent. In contrast, for the combination studies in Figure 7, we intentionally used nutlin-3a at concentrations lower than the IC50 values in growth inhibitory study (4 miroM of nutlin-3a for NCI-H28, 1 microM for MSTO-211H and 2 microM for NCI-H226). All the nutlin-3a concentrations used in Figure 7 were thus lower than those in Figure 3B. Please see the Figure 7 legend in line 671-674 page 41-42, which mentioned the concentrations in detail. We also explained the reason for different concentrations used in Figure 7 in the legend in page 669-671 in page 41. In that sense, the situation is the same as p-H2AX in Figure 3B and 4B, which used different exposure times.

(b) Rationale for Figure 7: We selected these lower doses to avoid a "ceiling effect," where a high dose of nutlin-3a alone might saturate the p53 response, thereby masking any synergistic or additive effects when combined with defactinib. Consequently, the up-regulation of p-p53 in Figure 7 appears less pronounced than in Figure 3B.

(c) Mechanism of up-regulation: While nutlin-3a increases p53 levels by inhibiting MDM2-mediated degradation, RITA can further enhance p53 phosphorylation through the induction of DNA damage and subsequent kinase activation. Furthermore, many factors influence the augmentation levels of p53 in cells treated with MDM2 inhibitors such as stability and affinity of the inhibitors, and susceptibility to ubiquitination in respective cells. The expression levels of p-p53 are therefore the outcomes of these factors involved. As shown in Supplementary Tables 5 (nutlin-3a) and 6 (RITA), the phosphorylated p53 levels varies depending on the cells and the inhibitor used. Phosphorylated p53 levels mediated by nutlin-3a were not always similar to those by RITA.

Comment (4) 4 Drugs and cell lines. IC50 values in 1-6 micromolar range is not so good actually, effects vary dramatically between cell lines (though expected). EHMES-10 resistant to both MDM2 inhibitors it looks like and NCI-H226 shows unusual responses in my opinion. Variability limits generalizability and clinical translation.

[Reply]

Thank you very much for your thoughtful comment regarding the drug sensitivities and clinical translation. We agree that the observed variability and the micromolar-range IC50 values are critical points.

(a) Variability and TP53 genotype: A key finding of this study is that sensitivity of nutlin-3a was dependent on the TP53 genotype. We showed that wild-type TP53 cells were sensitive, whereas mutated or non-functional p53 cells (NCI-H2452, Met-5A cells) were resistant (line 234-236 in page 15). This suggests that the INK4A/ARF deletion with wild-type TP53, often found in the majority of mesothelioma, could serve as a biomarker for MDM2 inhibitor sensitivity. We presume that identifying such "resistant" and "sensitive" profiles is not a limitation but rather a step toward patient stratification and personalized medicine. In contrast, sensitivity to defectinib was not linked with the TP53 genotype and sensitivity range of defectinib (Figure 2A) was not as great as that of nutlin-3a (in Figure 3A). Furthermore, sensitivity to RITA was not linked with the TP53 genotype but can be influenced by induction of DNA damage responses. Drug sensitivity is influenced by many factors such as genetic alterations and modifications of the target proteins; consequently, the sensitivity will be variable depending on properties of inhibitors and target cells. This study indicated that the sensitivity was subjected the agents and cells tested although the TP53 genotype can play a role in the sensitivity.

(b) EHMES-10 NCI-H226 cells: As the reviewer noted, EHMES-10 cells exhibited resistance to both inhibitors despite being wild-type TP53. We mentioned this in the Results section and suggested that a defect in downstream p53 pathways (e.g., impaired p21 induction) may result in reduced cell cycle arrest and contribute to this phenotype. We modified the sentence to explain more (lines 240–242 in page 15). Regarding NCI-H226, its response to nutlin-3a was relatively consistent with other wild-type TP53 cells; however, we acknowledge that differences in the RITA sensitivity probably due to some kinds of protein modification levels or other compensatory pathways, both of which might influence DNA damage responses and inhibition of MDM2 activity.

(c) Clinical translation: We acknowledge that the potency of the inhibitors used in this study will be lower than that of current small-molecule inhibitors that operate in the nanomolar range. We agree that high-dose applications increase the risk of off-target effects. This study however serves as a preclinical proof-of-concept to explore the synergy between p53 stabilization and FAK inhibition. These findings provide a rationale for developing next-generation agents with higher affinity and better pharmacological profiles. We revised the Discussion section and explained more (line 417-421 and line 422-423 in page 26). We also added reference 40 for idasanutlin which was mentioned by the reviewer.

(d) Future drug development: We would like to contend that "negative" results or variability are informative. Our data suggest that while this combination is promising for epithelioid type mesothelioma with wild-type TP53 genotype, different strategies will be required for sarcomatoid or biphasic types which often have mutated TP53 genotype (line 407-411 in page 25-26). We believe that variability of drug sensitivity will not be negative in terms of clinical applications but rather stimulate further investigations. We mentioned these in Discussion as above.

Comment (5) 5 Direct comparisons difficult since only one exposure duration tested.

[Reply]

Thank you very much for your valuab

---

## [Decision Letter · Decision Letter 2]

8 Feb 2026

Strategic selection of MDM2 inhibitors enhances the efficacy of FAK inhibition in mesothelioma based on TP53 genotype

PONE-D-25-46063R2

Dear Dr. Masatoshi,

We’re pleased to inform you that your manuscript has been judged scientifically suitable for publication and will be formally accepted for publication once it meets all outstanding technical requirements.

Kind regards,

Zu Ye, Ph.D.

Academic Editor

PLOS One

Additional Editor Comments (optional):

Reviewers' comments:

Reviewer's Responses to Questions

**Comments to the Author**

Reviewer #3: All comments have been addressed

2. Is the manuscript technically sound, and do the data support the conclusions?

Reviewer #3: Yes

3. Has the statistical analysis been performed appropriately and rigorously?

Reviewer #3: I Don't Know

4. Have the authors made all data underlying the findings in their manuscript fully available?

Reviewer #3: Yes

5. Is the manuscript presented in an intelligible fashion and written in standard English?

Reviewer #3: Yes

Reviewer #3: The authors have made significant efforts to address this referees concerns, particularly regarding technical errors in data labeling and clarity. There remains a few editing issues to look into (spelling errors, drug name needs to be correct, typos and so on).

**Do you want your identity to be public for this peer review?** For information about this choice, including consent withdrawal, please see our Privacy Policy

Reviewer #3: No

---

## [Editor Report · Acceptance letter]

PONE-D-25-46063R2

PLOS One

Dear Dr. Tagawa,

I'm pleased to inform you that your manuscript has been deemed suitable for publication in PLOS One. Congratulations! Your manuscript is now being handed over to our production team.

Kind regards,

on behalf of

Prof. Zu Ye

Academic Editor

PLOS One